# Multidrug-resistant *Klebsiella pneumoniae* coinfection with multiple microbes: a retrospective study on its risk factors and clinical outcomes

Xixi Song,[1] Chonghe Xu,[2,3] Zhongqi Zhu,[1] Chenchen Zhang,[1] Chao Qin,[1] Juan Liu,[1] Xiaoli Kong,[1] Zhijun Zhu,[1] Wei Xu,[4] Mei Zhu[1]

**ABSTRACT** The prevalence of multidrug-resistant *Klebsiella pneumoniae* (MDR-KP) is rising globally. The aim of this study was to investigate the epidemiology, risk factors, and clinical outcomes of MDR-KP coinfections with multiple microbes and infections with carbapenem-resistant *Klebsiella pneumoniae* (CRKP) among patients in a tertiary hospital in China and to establish an individualized linear prediction model. In this retrospective study, patients admitted from January 2021 to March 2024 with a diagnosis of MDR-KP infection were included. We recorded demographics, comorbidities, laboratory indicators, therapeutic interventions, antimicrobial susceptibility tests (ASTs), and analyzed clinical outcomes. Logistic regression models were employed to evaluate the risk factors associated with MDR-KP coinfections and infections with CRKP. A total of 164 patients with MDR-KP infection were included. Of these patients, 78 (47.6%) were infected with MDR-KP only, and 86 (52.4%) were coinfected with other microbes; 115 (70.1%) were infected with extended-spectrum beta-lactamase-producing *Klebsiella pneumoniae* (ESBL-KP), and 49 (29.9%) were infected with CRKP. The most common source of infection among patients with MDR-KP infection was the respiratory tract (96/164, 58.5%), followed by the urinary tract (31/164, 18.9%). Multivariate logistic regression analysis showed that nasogastric catheters (odds ratio [OR], 5.351; 95% confidence inteval [CI], 1.437–19.926, $P = 0.012$), as well as venous and arterial catheters (OR, 5.182; 95% CI, 1.272–21.113, $P = 0.022$), were independent risk factors for coinfection. The total risk score for all factors was 143.3, with a predicted risk rate ranging from 0.25 to 0.85. In the receiver operating characteristic (ROC) analysis, the area under the curve (AUC) for predicting coinfection based on the total risk score was 0.773 (95% CI: 0.7054–0.8405). Tracheostomy (OR, 4.673; 95% CI, 1.153–18.937, $P = 0.031$) and fiberoptic bronchoscopy (OR, 4.041; 95% CI, 1.305–12.516, $P = 0.015$) were independent risk factors for infection with CRKP. The total risk score for all factors was 193.9, with a predicted risk rate ranging from 0.15 to 0.85. In the ROC analysis, the AUC for predicting CRKP using the total risk score was 0.752 (95% CI: 0.6739–0.8306). Analysis of the calibration curve indicated good agreement between the observed and predicted values. The log-rank test was used to compare all-cause mortality between the two groups, and 30-day mortality was higher in the coinfected group than that in the MDR-KP alone group ($P = 0.03$). There was no significant difference in 30-day mortality between the CRKP and ESBL-KP groups ($P = 0.09$). This study successfully established a predictive model based on risk factors, which has good predictive value for both patients with coinfections and those with CRKP. Coinfections and CRKP infections were significantly associated with increased overall mortality, elevated healthcare costs, and poor prognosis in patients. These findings provided a basis for further clinical research and optimization of management strategies for MDR-KP coinfections and CRKP infections.

**Peer Reviewers** Yuanzhao Ding, University of Oxford, Oxford, United Kingdom; Nana Ama Amissah, Noguchi Memorial Institute for Medical Research, Legon, Accra, Ghana; Mohamed Abdelmonem, Stanford Healthcare Valley Care, Pleasanton, California, USA

Address correspondence to Wei Xu, weixu532@sina.com, xuwei@ahmu.edu.cn, or Mei Zhu, zhumei@ahmu.edu.cn, meizhu532@sina.com.

Xixi Song, Chonghe Xu, Zhongqi Zhu, and Chenchen Zhang contributed equally to this article. The author order was determined alphabetically by family name.

The authors declare no conflict of interest.

See the funding table on p. 19.

**IMPORTANCE** Coinfections and carbapenem-resistant *Klebsiella pneumoniae* (CRKP) infections significantly increased morbidity and economic burden, leading to longer intensive care unit (ICU) stays and poorer prognoses. Coinfection may also lead to a higher 30-day mortality rate. Patients suffering from multidrug-resistant *Klebsiella pneumoniae* (MDR-KP) used two or more antibiotics for infection control, but the therapeutic outcomes remained suboptimal. In order to reverse the rising trend in mortality rate associated with coinfection and CRKP infection, certain measures need to be taken: (i) develop stricter protocols for terminal cleaning of rooms (especially ICUs), cleaning of equipment (such as bronchoscopes) and hand hygiene; (ii) conduct antimicrobial resistance gene testing in the healthcare environment and implement antimicrobial stewardship to optimize antibiotic consumption and reduce the emergence and spread of multidrug-resistant organisms.

**KEYWORDS** multidrug resistance, coinfection, *Klebsiella pneumoniae*, carbapenem resistance, prediction model, mortality

*K*lebsiella pneumoniae (KP), a member of the Enterobacteriaceae family, is an opportunistic pathogen that accounts for approximately one-third of all Gram-negative infections (1). It is included in the 2024 WHO Bacterial Priority Pathogens List (WHO BPPL) as well as ESKAPE pathogens (2), and is responsible for a wide range of infections, including urinary tract infections, pneumonia, bacteremia, and liver abscesses (3). Antibiotic resistance (AR) is a complex process involving multiple factors (4). For example, the selective pressure posed by the extensive use of antibiotics has facilitated the emergence of multidrug-resistant *Klebsiella pneumoniae* (MDR-KP). Moreover, the conjugative transfer of antibiotic resistance genes between bacterial species and genera has exacerbated the problem of antibiotic resistance (5). A systematic review and meta-analysis showed that the prevalence of nosocomial infections caused by MDR-KP and extended-spectrum beta-lactamase-producing *K. pneumoniae* (ESBL-KP) in South-Eastern Asia was estimated to be 55% (95% confidence interval [CI], 9–96) and 27% (95% CI, 32–100), respectively (6). In fact, multidrug-resistant bacteria are more likely to be found among immunocompromised patients, featuring high mortality, reduced effectiveness of drugs, prolonged hospitalization, and high medical cost (7). Therefore, infection with MDR-KP is a thorny issue for global public health.

The incidence of polymicrobial infections, defined as infections that involve two or more types of bacteria, viruses, fungi, or parasites, has been rising for the past decades. Although polymicrobial infections are increasingly prevalent among critically ill patients, the major MDR infection control strategies and antibiotic regimens are commonly based on the assumption of a single-microbe infection (8). Poor wound care, inappropriate use of antibiotics, and poor hospital hygienic management could be the common causes of polymicrobial infections (9, 10). Polymicrobial infections are often associated with a poorer prognosis, including prolonged hospital and intensive care unit stays, increased incidence of septic shock, higher demand for amputation due to diabetic ulcers, and increased all-cause mortality (11–13).

In an environment of rising multidrug-resistant infection rates, research on the risk factors and outcomes of multi-microbial infections is still limited and restricted to a single population of patients with bloodstream infection or diabetic foot (10, 11). Given that polymicrobial infections can be identified in a variety of populations, this limitation may confound the proper perception of polymicrobial infections among clinical healthcare professionals, and few studies have focused on polymicrobial infections involving MDR-KP. Hence, we tried to clarify the risk factors of polymicrobial infections involving MDR-KP, establish a predictive model, and promote the prevention and control of carbapenem-resistant *Klebsiella pneumoniae* (CRKP) infection.

## MATERIALS AND METHODS

### Patients and grouping

This study retrospectively included 164 patients infected with MDR-KP at a tertiary hospital in China from January 2022 to March 2024. The inclusion criteria were as follows: (i) hospitalized patients; (ii) patients with a confirmed infection caused by MDR-KP alone or in combination with other pathogens; and (iii) availability of complete medical records. The exclusion criteria were as follows: (i) duplicate positive cultures; (ii) repeat hospitalizations; (iii) patients still hospitalized at the time of data collection; and (iv) patients with key data missing. In our study, infections with MDR-KP were divided into two groups according to the category of infected microbes: MDR-KP alone and coinfected with other microbes. In addition, patients were also classified into two groups based on their resistance phenotype: ESBL-KP group and CRKP group.

### Clinical data

Relevant clinical variables were collected from the hospital's electronic medical records. The data included age, gender, admission and discharge diagnoses, comorbidities (hypertension, diabetes mellitus, respiratory disease, digestive system disease, cerebro-vascular disease, cardiovascular disease, chronic kidney disease, chronic liver disease, and solid tumor), age-adjusted Charlson comorbidity index (aCCI), quick sequential organ failure assessment (qSOFA), activity of daily living (ADL), past medical history, invasive operations, therapeutic drugs, source of infection, antibiotic susceptibility results, laboratory examination results at admission, admission to intensive care unit (ICU), length of stay (LOS), and clinical outcomes.

### Variable definitions

MDR was defined as acquired resistance to at least one agent in three or more antimicrobial drug categories (14). Coinfection referred to an infection in which MDR-KP and other microbes were isolated from the same or different clinical specimens through the laboratory within seven days. According to the 2021 guidelines from the Clinical and Laboratory Standards Institute (CLSI), carbapenem resistance in bacterial strains was defined as an MIC ≥4 mg/L for meropenem/imipenem or ≥2 mg/L for ertapenem via broth microdilution (15). Patients were categorized as immunocompetent or immuno-suppressed according to established criteria (16). qSOFA was used for rapid screening of sepsis (17). Upon admission, the functional status of patients was evaluated using the ADL index (18), which measures their capacity to carry out activities of daily living independently. Patients' overall systemic health was assessed by aCCI (19). Administration of antibiotics prior to AST was defined as empirical treatment and otherwise as definitive treatment. Combined antibiotic therapy included no less than two types of antibiotics and lasted for more than 24 h. The length of post-infection hospitalization was defined as the time from the date of the first collection when the sample was tested positive to discharge. In this study, the 30- or 90-day mortality was defined as all-cause deaths within 30 or 90 days after infection.

### Microbiological analysis

In our hospital, all isolates were identified by matrix-assisted laser desorption/ionization time-of-flight mass spectrometry (MALDI-TOF-MS; Bruker Corporation, Karlsruhe, Germany). According to the breakpoints for Enterobacteriaceae set by CLSI 2021 guidelines, AST results were determined by MIC values obtained by the dilution method and the diameters of the inhibition circle obtained by the Kirby–Bauer's disk diffusion (KB) method. Sixteen antimicrobial agents were tested: amikacin, imipenem, piper-acillin/tazobactam, cefotetan, gentamicin, cefepime, sulfamethoxazole-trimethoprim, ceftazidime, levofloxacin, tobramycin, aztreonam, ciprofloxacin, ampicillin/sulbactam, ceftriaxone, cefazolin, and ampicillin. Strains were considered sensitive or non-sensitive

(either intermediate or resistant) to each antibiotic tested. *Pseudomonas aeruginosa* ATCC 27853 and *Escherichia coli* ATCC 25922 were used as quality control strains in this experiment.

## Statistical analysis

Variables were analyzed using IBM statistical product and service solutions (SPSS) (version 25.0). Categorical variables were expressed as frequencies and percentages, while continuous variables were expressed as the mean ± standard deviation for normally distributed data or as the interquartile range (IQR) for those not normally distributed. Continuous variables that met the normal distribution were analyzed with a two-independent sample *t*-test, while the Mann-Whitney *U* test was used for continuous variables conforming to skewed distribution. Categorical variables were tested using the χ test or Fisher's exact test, as appropriate. Variables with $P < 0.05$ in univariate analysis were subjected to a multicollinearity assessment using the variance inflation factor (VIF), and those with a VIF cut-off higher than 10 were excluded. The final multivariable logistic regression model included variables without multicollinearity and adjusted for gender and age as confounders. Multivariable logistic regression analysis identified risk factors associated with MDR-KP coinfection with multiple microbes and CRKP, expressed as odds ratios (ORs) and 95% confidence intervals (CIs). The statistical tests were analyzed using two-tailed tests, and a *P* value less than 0.05 was considered statistically significant. We used R4.1.1 software (R Foundation, Austria) to establish the prediction model of the nomogram. The caret package was applied for internal validation via the bootstrap method, while the pROC package calculated the AUC to assess the discriminative ability of the model. Model calibration was assessed using a calibration plot. The Kaplan-Meier method was used to evaluate 30-day survival, and the log-rank test was used to compare survival curves.

## RESULTS

### Demographic clinical characteristics of patients

The study included 164 patients who were first hospitalized for infection with MDR-KP between January 2022 and March 2024. Of these patients, 78 (47.6%) were infected with MDR-KP alone, and 86 (52.4%) were coinfected with other microbes. Among the 164 patients enrolled, 115 (70.1%) were infected with extended-spectrum beta-lactamase-producing *Klebsiella pneumoniae*, and 49 (29.9%) were infected with carbapenem-resist-ant *Klebsiella pneumoniae* (Fig. 1).

Detailed demographic information and clinical characteristics of the 164 patients were presented in Tables 1 and 2. Infection with MDR-KP occurred more frequently among men (129/164, 78.7%). The study population was predominantly elderly, with a high proportion of patients aged ≥65 (112/164, 68.3%). Only 18.3% (30/164) of the isolates were healthcare-associated, while 28.0% (46/164) were community-acquired, and 53.7% (88/164) were hospital-acquired. Strains of the MDR-KP only group were common in hospital-acquired (32/78, 41.0%) and community-acquired (31/78, 39.7%) cases. The strains of the coinfection group were predominantly hospital-acquired, and this proportion was significantly larger than that in the MDR-KP only group (65.1% vs 41.0%, $P = 0.002$) (Table 1). Strains of the CRKP group were prevalent in hospital-acquired cases (31/49, 63.3%). More strains of the ESBL-KP group were obtained in the community compared with the CRKP group (Table 2). Coinfected patients featured a greater proportion of smoking (12.8% vs 10.3%, $P = 0.613$), alcohol consumption (11.6% vs 6.4%, $P = 0.247$), prior surgical history (19.8% vs 10.3%, $P = 0.091$), and prior hospitalization history (48.8% vs 39.7%, $P = 0.242$). Compared with the MDR-KP only group, patients in the coinfection group were more likely to have pneumonia (50.0% vs 34.6%, $P = 0.047$) within 3 months (Table 1). Compared with the ESBL-KP group, the CRKP group had a higher proportion of prior surgical history (24.5% vs 11.3%, $P = 0.032$) within 3 months (Table 2), and their aCCI scores were higher (median, 5 vs 4, $P = 0.168$) in the coinfection

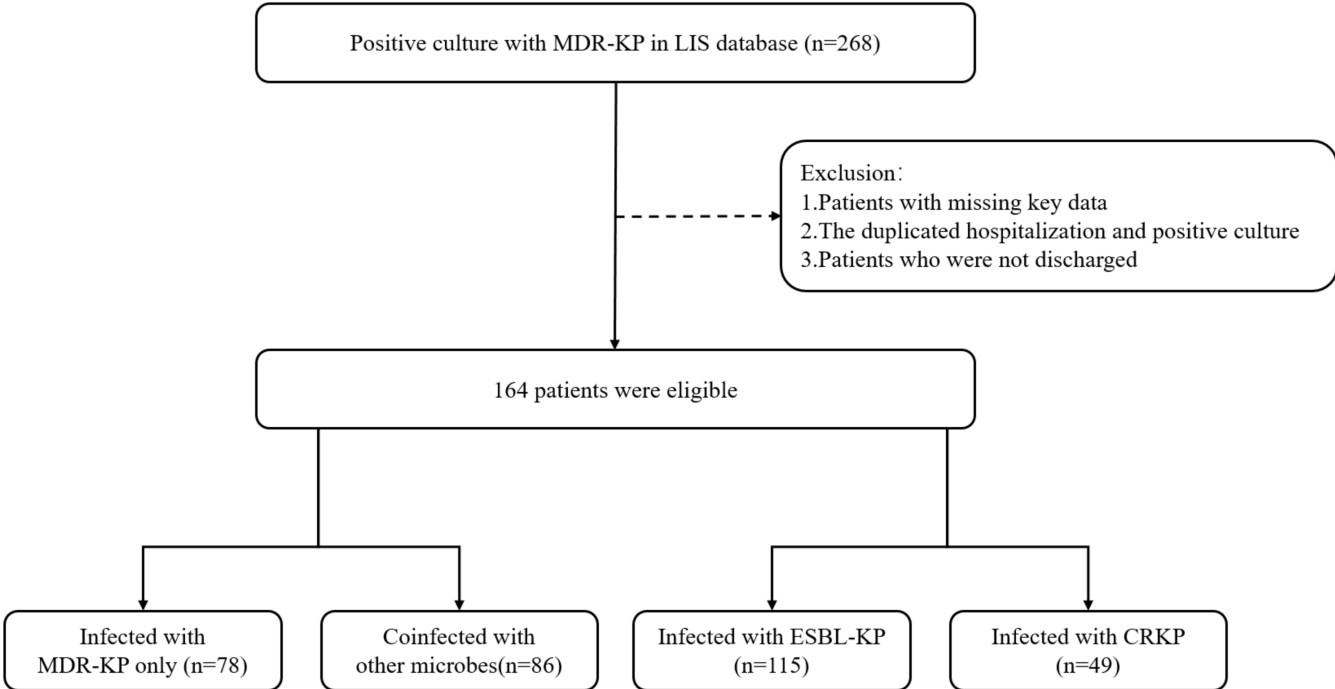

FIG 1 Flowchart for enrolled patients.

group, although not statistically significant. The patients' ability to perform activities of daily living, as measured by the ADL score at admission, was significantly lower in the coinfection group (median, 20 vs 47.5, $P = 0.010$) and CRKP group (median, 10 vs 45, $P < 0.001$) (Tables 1 and 2). The qSOFA ≥2 implied the possibility of organ dysfunction: a higher proportion of patients in the coinfection group on the day of admission had a qSOFA score ≥2 (18.6% vs 9.0%, $P = 0.003$) (Table 1), and a higher proportion of patients in the CRKP group had a qSOFA score ≥2 compared with the ESBL-KP group (24.5% vs 9.6%, $P = 0.012$) (Table 2).

Most patients (152/164, 92.7%) had comorbidities, with respiratory disease (107/164, 65.2%) being the most commonly reported, followed by hypertension (77/164, 47.0%), cerebrovascular disease (70/164, 42.7%), and cardiovascular disease (48/164, 29.3%). There were significant differences in respiratory disease and cerebrovascular disease (76.7% vs 52.6% and 52.3% vs 32.1%, $P = 0.001$ and $P = 0.009$, respectively) between the coinfection group and MDR-KP only group (Table 1). Cerebrovascular disease was more likely to occur in the CRKP group compared with the ESBL-KP group (57.1% vs 36.5%, $P = 0.015$). Additionally, patients infected with ESBL-KP were at greater risk of developing solid organ tumors (22.6% vs 8.2%, $P = 0.029$) (Table 2).

To be noted, the vast majority of patients (146/164, 89.0%) had undergone invasive procedures during their hospitalizations. Urinary catheter (75.6% vs 47.4%, $P < 0.001$), nasogastric catheter (65.1% vs 16.7%, $P < 0.001$), venous and arterial catheter (48.8% vs 17.9%, $P < 0.001$), mechanical ventilation (45.3% vs 21.8%, $P = 0.001$), tracheal cannula (51.2% vs 19.2%, $P < 0.001$), tracheostomy (26.7% vs 5.1%, $P < 0.001$), and fiberoptic bronchoscopy (41.9% vs 14.1%, $P < 0.001$) were more likely to be used in the coinfection group compared with the MDR-KP only group (Table 1). The probability of receiving nasogastric catheter (61.2% vs 33.9%, $P = 0.001$), venous and arterial catheter (53.1% vs 26.1%, $P = 0.001$), mechanical ventilation (59.2% vs 23.5%, $P < 0.001$), tracheal cannula (57.1% vs 27.0%, $P < 0.001$), tracheostomy (36.7% vs 7.8%, $P < 0.001$), and fiberoptic bronchoscopy (55.1% vs 17.4%, $P < 0.001$) was greater in the CRKP group compared with the ESBL-KP group (Table 2). Patients in the coinfection and CRKP groups spent more in the hospital, indicating a heavier financial burden caused by the infection. The

**TABLE 1** Comparison of clinical characteristics in MDR-KP only and coinfection groups[c]

| Variable | Total (*n* = 164) | MDR-KP only (*n* = 78) | Coinfection (*n* = 86) | *P*-value[a] |
|---|---|---|---|---|
| Sex (male) | 129 (78.7%) | 64 (82.1%) | 65 (75.6%) | 0.313 |
| Age group, years | | | | |
| ≤35 | 7 (4.3%) | 4 (5.1%) | 3 (3.5%) | 0.604 |
| 36–64 | 45 (27.4%) | 20 (25.6%) | 25 (29.1%) | 0.623 |
| ≥65 | 112 (68.3%) | 54 (69.2%) | 58 (67.4%) | 0.806 |
| Infection type | | | | |
| Community-acquired | 46 (28.0%) | 31 (39.7%) | 15 (17.4%) | **0.001** |
| Healthcare-associated | 30 (18.3%) | 15 (19.2%) | 15 (17.4%) | 0.767 |
| Hospital-acquired | 88 (53.7%) | 32 (41.0%) | 56 (65.1%) | **0.002** |
| Past history within 3 months | | | | |
| Smoking | 19 (11.6%) | 8 (10.3%) | 11 (12.8%) | 0.613 |
| Alcohol consumption | 15 (9.1%) | 5 (6.4%) | 10 (11.6%) | 0.274 |
| Surgical history | 25 (15.2%) | 8 (10.3%) | 17 (19.8%) | 0.091 |
| Hospitalization history | 73 (44.5%) | 31 (39.7%) | 42 (48.8%) | 0.242 |
| Prior infection history within 3 months | | | | |
| Pneumonia | 70 (42.7%) | 27 (34.6%) | 43 (50.0%) | **0.047** |
| Hepatitis | 66 (40.2%) | 32 (41.0%) | 34 (39.5%) | 0.846 |
| Urinary tract infection | 27 (16.5%) | 15 (19.2%) | 12 (14.0%) | 0.363 |
| Skin and soft tissue infection | 7 (4.3%) | 2 (2.6%) | 5 (5.8%) | 0.295 |
| Bloodstream infection | 8 (4.9%) | 5 (6.4%) | 3 (3.5%) | 0.295 |
| Fever ≥ 72 h | 29 (17.7%) | 9 (11.5%) | 20 (23.3%) | 0.050 |
| aCCI | 4 (3, 5) | 4 (3, 5) | 5 (3, 6) | 0.168 |
| ADL score at admission | 35 (10, 70) | 47.5 (16.3, 80.0) | 20 (6.3, 48.8) | **0.010** |
| qSOFA ≥ 2 at admission | 23 (14.0%) | 7 (9.0%) | 16 (18.6%) | **0.003** |
| Comorbidities | | | | |
| Hypertension | 77 (47.0%) | 34 (43.6%) | 43 (50.0%) | 0.411 |
| Diabetes mellitus | 35 (21.3%) | 16 (20.5%) | 19 (22.1%) | 0.805 |
| Respiratory disease | 107 (65.2%) | 41 (52.6%) | 66 (76.7%) | **0.001** |
| Digestive system disease | 36 (22.0%) | 18 (23.1%) | 18 (20.9%) | 0.740 |
| Cerebrovascular disease | 70 (42.7%) | 25 (32.1%) | 45 (52.3%) | **0.009** |
| Cardiovascular disease | 48 (29.3%) | 19 (24.4%) | 29 (33.7%) | 0.188 |
| Chronic kidney disease | 41 (25.0%) | 17 (21.8%) | 24 (27.9%) | 0.367 |
| Chronic liver disease | 31 (18.9%) | 15 (19.2%) | 16 (18.6%) | 0.919 |
| Solid tumor | 30 (18.3%) | 11 (14.1%) | 19 (22.1%) | 0.186 |
| Immunosuppression | 105 (64.0%) | 45 (57.7%) | 60 (69.8%) | 0.108 |
| Invasive procedure | | | | |
| Urinary catheter | 102 (62.2%) | 37 (47.4%) | 65 (75.6%) | **<0.001** |
| Nasogastric catheter | 69 (42.1%) | 13 (16.7%) | 56 (65.1%) | **<0.001** |
| T-tube intubation | 7 (4.3%) | 3 (3.8%) | 4 (4.7%) | 0.799 |
| Venous and arterial catheter | 56 (34.1%) | 14 (17.9%) | 42 (48.8%) | **<0.001** |
| Mechanical ventilation | 56 (34.1%) | 17 (21.8%) | 39 (45.3%) | **0.001** |
| Tracheal cannula | 59 (36.0%) | 15 (19.2%) | 44 (51.2%) | **<0.001** |
| Tracheostomy | 27 (16.5%) | 4 (5.1%) | 23 (26.7%) | **<0.001** |
| Endoscope | 19 (11.6%) | 7 (9.0%) | 12 (14.0%) | 0.320 |
| Fiberoptic bronchoscopy | 47 (28.7%) | 11 (14.1%) | 36 (41.9%) | **<0.001** |
| Others | 38 (23.2%) | 18 (23.1%) | 20 (23.3%) | 0.978 |
| Related cost (US dollars)[b] | | | | |
| Total cost | 3,371.3 (1,506.8, 7,458.2) | 1,142.4 (7,458.2, 3,371.3) | 4,771.0 (3,371.3, 1,142.4) | **<0.001** |
| Medical fee | 855.0 (370.0, 2,144.1) | 280.8 (2,144.1, 855.0) | 1,485.1 (855.0, 280.8) | **<0.001** |
| Antibacterial agent cost | 212.7 (97.4, 559.9) | 78.2 (559.9, 212.7) | 419.1 (212.7, 78.2) | 0.889 |
| Laboratory findings at admission | | | | |
| WBC, ×10⁹/L | 9.04 (6.00, 13.03) | 9.2 (5.89, 13.74) | 8.71 (6.03, 11.69) | 0.963 |

*(Continued on next page)*

TABLE 1 Comparison of clinical characteristics in MDR-KP only and coinfection groups[c] (*Continued*)

| Variable | Total (*n* = 164) | MDR-KP only (*n* = 78) | Coinfection (*n* = 86) | *P*-value[a] |
|---|---|---|---|---|
| Neu, ×10$^9$/L | 7.02 (4.17, 10.76) | 6.78 (4.10, 11.55) | 7.13 (4.27, 9.97) | 0.979 |
| Lym, ×10$^9$/L | 1.07 (0.63, 1.55) | 0.91 (0.63, 1.42) | 1.22 (0.60, 1.71) | 0.071 |
| Mon, ×10$^9$/L | 0.51 (0.33, 0.71) | 0.51 (0.33, 0.69) | 0.51 (0.33, 0.76) | 0.974 |
| PLT, ×10$^9$/L | 182 (129, 233) | 181 (118, 216) | 182 (139, 255) | 0.113 |
| Hb, g/L | 118 (98, 131) | 122 (105, 133) | 115 (92, 128) | 0.138 |
| NLR | 7.17 (3.46, 14.71) | 7.25 (3.76, 15.46) | 6.92 (3.26, 13.59) | 0.800 |
| PLR | 179.28 (118.01, 266.02) | 183.88 (120.26, 261.71) | 175.70 (109.66, 271.14) | 0.795 |
| CAR | 0.46 (0.01, 2.51) | 0.46 (0, 3.15) | 0.48 (0.02, 2.15) | 0.629 |
| CALLY index | 0.20 (0.03, 1.74) | 0.23 (0.02, 1.74) | 0.18 (0.04, 1.74) | **0.014** |
| ALT, U/L | 26 (16, 42) | 27 (16, 49) | 25 (17, 40) | 0.991 |
| AST, U/L | 28 (21, 47) | 27 (19, 55) | 31 (21, 46) | 0.290 |
| TB, μmol/L | 14.0 (9.0, 19.0) | 16.5 (11.1, 22.0) | 12.0 (8.7, 17.0) | **0.003** |
| TP, g/L | 63.6 ± 9.0 | 63.9 ± 8.5 | 63.3 ± 9.5 | 0.661 |
| ALB, g/L | 35.3 ± 6.4 | 36.0 ± 6.1 | 34.7 ± 6.6 | 0.210 |
| Cr, μmol/L | 72.0 (56.8, 106.3) | 72.5 (59.3, 99.8) | 70.5 (51.3, 107.8) | 0.407 |
| Cys C, mg/L | 1.18 (0.91, 1.71) | 1.21 (0.92, 1.92) | 1.17 (0.90, 1.62) | 0.488 |
| Bun, mmo/L | 7.4 (4.8, 11.0) | 7.6 (5.3, 11.0) | 6.9 (4.5, 10.8) | 0.197 |

[a]*P*-values were obtained using χ for categorical variables and Kruskal Wallis test for continuous variables.
[b]1 US dollar = 7.0978 Chinese Yuan as of March 2024.
[c]Data are represented as no. (%), interquartile range (IQR) and mean ± standard error of the mean (SEM). Bold values indicate statistical significance (*P* < 0.05). MDR-KP, multidrug-resistant *Klebsiella pneumoniae*; aCCI, age-adjusted Charlson comorbidity index; ADL, activity of daily living; qSOFA, quick sequential organ failure assessment; WBC, white blood cells; Neu, neutrophils; Lym, lymphocytes; Mon, monocytes; PLT, platelet counts; Hb, hemoglobin; NLR, neutrophil-to-lymphocyte ratio; PLR, platelet-to-lymphocyte ratio; CAR, C-reactive protein-to-albumin ratio; CALLY index, C-reactive protein-albumin-lymphocyte index; ALT, alanine aminotransferase; AST, aspartate aminotransferase; TB, total bilirubin; TP, total protein; ALB, albumin; Cr, creatinine; Cys C, cystatin C; Bun, blood urea nitrogen.

coinfection group showed a lower C-reactive protein-albumin-lymphocyte (CALLY) index compared with the MDR-KP only group (median, 0.18 vs 0.23, *P* = 0.014), indicating a higher death risk. The coinfection group also showed a deficiency in total bilirubin (median, 12.0 vs 16.5, *P* = 0.003) compared with the MDR-KP only group at the first laboratory examination (Table 1). Compared with the ESBL-KP group, the white blood cell count (median, 10.17 vs 8.31, *P* = 0.015), as well as the neutrophil count (median, 8.57 vs 6.49, *P* = 0.007), was higher in the CRKP group. The neutrophil-to-lymphocyte ratio (NLR), an important indicator for assessing the inflammatory response, was higher in the CRKP group compared with the ESBL-KP group (median, 9.63 vs 5.64, *P* = 0.018). The C-reactive protein-to-albumin ratio (CAR) is a new composite index reflecting the inflammatory and nutritional status. In our study, the CRKP group showed a higher level of CAR (median, 1.22 vs 0.33, *P* = 0.013) and a lower protein synthesis capacity (total protein and albumin) (Table 2), indicating a poor prognosis.

## Microbiological characteristics of patients

The isolation sites of the MDR-KP strains were analyzed (Fig. 2A). The most common source was the respiratory tract (96/164, 58.5%), followed by the urinary tract (31/164, 18.9%) and bloodstream infection (12/164, 7.3%), etc. Among patients coinfected with only one type of bacterium (Fig. 2B), *Escherichia coli* had the largest proportion of infections (12/86, 14.0%), followed by *Acinetobacter baumannii* (11/86, 12.8%) and *Pseudomonas aeruginosa* (8/86, 9.3%). Coinfections with Gram-positive bacteria had also been recorded, which included *Staphylococcus aureus* (7/86, 8.1%) and *Enterococcus faecium* (2/86, 2.3%). The coinfected fungi were all *Candida species* (11/86, 12.8%). Furthermore, 19 patients (22.1%) were coinfected with more than one microbe.

Sixteen antibiotics were used for the antimicrobial susceptibility testing of 164 MDR-KP isolates. Almost all strains in both groups were resistant to ampicillin (98.7% vs 100.0%). Apart from cefotetan (24.4% vs 26.7%, *P* = 0.628), both MDR-KP only and coinfection groups had high rates of resistance to the cephalosporin antibiotics tested, such as cefazolin (97.4% vs 94.2%, *P* = 0.605), ceftriaxone (94.9% vs 93.0%, *P* = 0.621),

**TABLE 2** Comparison of clinical characteristics in ESBL-KP and CRKP groups[c]

| Variable | ESBL-KP (n = 115) | CRKP (n = 49) | P-value[a] |
|---|---|---|---|
| Sex (male) | 90 (78.3%) | 39 (79.6%) | 0.849 |
| Age group, years | | | |
| ≤35 | 4 (3.5%) | 3 (6.1%) | 0.457 |
| 36–64 | 34 (29.6%) | 11 (22.4%) | 0.350 |
| ≥65 | 77 (67.0%) | 35 (71.4%) | 0.573 |
| Infection type | | | |
| Community-acquired | 40 (34.8%) | 6 (12.2%) | **0.003** |
| Healthcare-associated | 18 (15.7%) | 12 (24.5%) | 0.180 |
| Hospital-acquired | 57 (49.6%) | 31 (63.3%) | 0.107 |
| Past history in 3 months | | | |
| Smoking | 12 (10.4%) | 7 (14.3%) | 0.481 |
| Alcohol consumption | 12 (10.4%) | 3 (6.1%) | 0.364 |
| Surgical history | 13 (11.3%) | 12 (24.5%) | **0.032** |
| Hospitalization history | 49 (42.6%) | 24 (49%) | 0.452 |
| Prior infection history within 3 months | | | |
| Pneumonia | 46 (40.0%) | 24 (49.0%) | 0.287 |
| Hepatitis | 45 (39.1%) | 21 (42.9%) | 0.656 |
| Urinary tract infection | 21 (18.3%) | 6 (12.2%) | 0.342 |
| Skin and soft tissue infection | 6 (5.2%) | 1 (2.0%) | 0.325 |
| Bloodstream infection | 5 (4.3%) | 3 (6.1%) | 0.636 |
| Fever ≥ 72 h | 17 (14.8%) | 12 (24.5%) | 0.136 |
| aCCI | 4 (3, 6) | 4 (3, 5) | 0.191 |
| ADL score at admission | 45 (12.5, 77.5) | 10 (0, 30) | **<0.001** |
| qSOFA ≥ 2 at admission | 11 (9.6%) | 12 (24.5%) | **0.012** |
| Comorbidities | | | |
| Hypertension | 52 (45.2%) | 25 (51.0%) | 0.496 |
| Diabetes mellitus | 23 (20.0%) | 12 (24.5%) | 0.521 |
| Respiratory disease | 71 (61.7%) | 36 (73.5%) | 0.149 |
| Digestive system disease | 25 (21.7%) | 11 (22.4%) | 0.920 |
| Cerebrovascular disease | 42 (36.5%) | 28 (57.1%) | **0.015** |
| Cardiovascular disease | 33 (28.7%) | 15 (30.6%) | 0.805 |
| Chronic kidney disease | 27 (23.5%) | 14 (28.6%) | 0.491 |
| Chronic liver disease | 22 (19.1%) | 9 (18.4%) | 0.909 |
| Solid tumor | 26 (22.6%) | 4 (8.2%) | **0.029** |
| Immunosuppression | 74 (64.3%) | 31 (63.3%) | 0.895 |
| Invasive procedure | | | |
| Urinary catheter | 66 (57.4%) | 36 (73.5%) | 0.052 |
| Nasogastric catheter | 39 (33.9%) | 30 (61.2%) | **0.001** |
| T-tube intubation | 5 (4.3%) | 2 (4.1%) | 0.938 |
| Venous and arterial catheter | 30 (26.1%) | 26 (53.1%) | **0.001** |
| Mechanical ventilation | 27 (23.5%) | 29 (59.2%) | **<0.001** |
| Tracheal cannula | 31 (27.0%) | 28 (57.1%) | **<0.001** |
| Tracheostomy | 9 (7.8%) | 18 (36.7%) | **<0.001** |
| Endoscope | 15 (13.0%) | 4 (8.2%) | 0.371 |
| Fiberoptic bronchoscopy | 20 (17.4%) | 27 (55.1%) | **<0.001** |
| Others | 24 (20.9%) | 14 (28.6%) | 0.285 |
| Related cost (US dollars)[b] | | | |
| Total cost | 2,622.1 (1,267.0, 6,870.2) | 4,492.0 (2,556.0, 10,759.7) | **0.001** |
| Medical fee | 785.3 (284.8, 1,830.9) | 1,306.4 (620.3, 3,027.3) | **0.004** |
| Antibacterial agent cost | 195.8 (88.0, 525.9) | 258.0 (117.7, 829.8) | 0.227 |
| Laboratory findings at admission | | | |
| WBC, ×10$^9$/L | 8.31 (5.75, 11.65) | 10.17 (7.15, 14.80) | **0.015** |

*(Continued on next page)*

TABLE 2 Comparison of clinical characteristics in ESBL-KP and CRKP groups[c] (*Continued*)

| Variable | ESBL-KP (*n* = 115) | CRKP (*n* = 49) | *P*-value[a] |
|---|---|---|---|
| Neu, ×10$^9$/L | 6.49 (3.96, 9.59) | 8.57 (5.91, 12.99) | **0.007** |
| Lym, ×10$^9$/L | 1.05 (0.65, 1.55) | 1.11 (0.54, 1.53) | 0.956 |
| Mon, ×10$^9$/L | 0.52 (0.34, 0.67) | 0.49 (0.33, 0.88) | 0.407 |
| PLT, ×10$^9$/L | 185 ± 79 | 197 ± 94 | 0.376 |
| Hb, g/L | 120 (103, 133) | 110 (85, 125) | **0.010** |
| NLR | 5.64 (3.19, 12.00) | 9.63 (5.16, 17.14) | **0.018** |
| PLR | 174.73 (119.52, 251.25) | 200.00 (107.69, 307.21) | 0.052 |
| CAR | 0.33 (0, 1.95) | 1.22 (0.10, 3.82) | **0.013** |
| CALLY index | 0.30 (0.04, 1.74) | 0.10 (0.02, 0.71) | 0.416 |
| ALT, U/L | 25 (17, 44) | 28 (15, 41) | 0.793 |
| AST, U/L | 27 (20, 47) | 34 (22, 47) | 0.159 |
| TB, μmol/L | 14.0 (9.6, 20.2) | 12.0 (9.0, 17.0) | 0.224 |
| TP, g/L | 64.9 ± 8.4 | 60.1 ± 9.7 | **0.003** |
| ALB, g/L | 36.2 ± 5.9 | 33.1 ± 7.1 | **0.006** |
| Cr, μmol/L | 72.0 (56.5, 105.5) | 76.0 (57.0, 107.0) | 0.931 |
| Cys C, mg/L | 1.19 (0.93, 1.70) | 1.14 (0.87, 1.69) | 0.430 |
| Bun, mmo/L | 7.4 (4.9, 10.4) | 7.8 (4.8, 12.2) | 0.372 |

[a]*P*-values were obtained using χ for categorical variables and Kruskal Wallis test for continuous variables.
[b]1 US dollar = 7.0978 Chinese Yuan as of March 2024.
[c]Data are represented as no. (%), interquartile range (IQR) and mean ± standard error of the mean (SEM). Bold values indicate statistical significance (*P* < 0.05). ESBL-KP, extended-spectrum beta-lactamase-producing *Klebsiella pneumoniae*; CRKP, carbapenem-resistant *Klebsiella pneumoniae*; aCCI, age-adjusted Charlson comorbidity index; ADL, activity of daily living; qSOFA, quick sequential organ failure assessment; WBC, white blood cells; Neu, neutrophils; Lym, lymphocytes; Mon, monocytes; PLT, platelet counts; Hb, hemoglobin; NLR, neutrophil-to-lymphocyte ratio; PLR, platelet-to-lymphocyte ratio; CAR, C-reactive protein-to-albumin ratio; CALLY index, C-reactive protein-albumin-lymphocyte index; ALT, alanine aminotransferase; AST, aspartate aminotransferase; TB, total bilirubin; TP, total protein; ALB, albumin; Cr, creatinine; Cys C, cystatin C; Bun, blood urea nitrogen.

ceftazidime (60.3% vs 66.3%, *P* = 0.566), and cefepime (50.0% vs 52.3%, *P* = 0.786). Both groups also had high rates of resistance to the quinolones tested, including ciprofloxacin (87.2% vs 80.2%, *P* = 0.976) and levofloxacin (67.9% vs 65.1%, *P* = 0.329). It was noteworthy that the coinfection group had a significantly higher resistance rate to amikacin than that of the MDR-KP only group (38.4% vs 17.9%, *P* = 0.003) (Fig. 2C). Additionally, 29.9% (49/164) of the MDR-KP isolates were carbapenem-resistant. The rate of resistance to sulfamethoxazole-trimethoprim (66.1% vs 28.6%, *P* < 0.001) was significantly higher in the ESBL-KP group than that in the CRKP group. The rates of resistance to ampicillin/sulbactam (88.7% vs 85.7%, *P* = 0.092), ceftriaxone (95.7% vs 87.8%, *P* = 0.159), and cefazolin (98.3% vs 89.5%, *P* = 0.060) were marginally higher in the ESBL group than those in the CRKP group. Except for the four antibiotics mentioned above, the CRKP group showed higher resistance to all tested antibiotics than that in the ESBL-KP group (Fig. 2D).

## Drug treatment and clinical outcomes

The use of antibiotics in MDR-KP patients during hospitalization was analyzed (Table 3). The majority of patients in both groups received empirical antibiotics (84.6% vs 84.9%, *P* = 0.962) for a similar duration of treatment (a median of 3 days) (Fig. 3A). Patients with coinfection more frequently received second- or third-generation cephalosporins (55.8% vs 46.2%, *P* = 0.216) and piperacillin-tazobactam (26.7% vs 25.6%, *P* = 0.873) as empirical treatments. Especially, the proportion of coinfected patients receiving two (25.6% vs 11.5%, *P* = 0.022) or three and more antibiotics (26.7% vs 10.3%, *P* = 0.007) was twice that of the MDR-KP only group. The duration of receiving combined antibiotic therapy was extended by 2 days in the coinfection group (median, 2 vs 0 day, *P* < 0.001) (Fig. 3E). Compare with the ESBL-KP group, patients in the CRKP group received empirical antibiotics for a slightly longer period, although it was not statistically significant (median, 4 vs 3 day, *P* = 0.804). The durations of treatment with single and

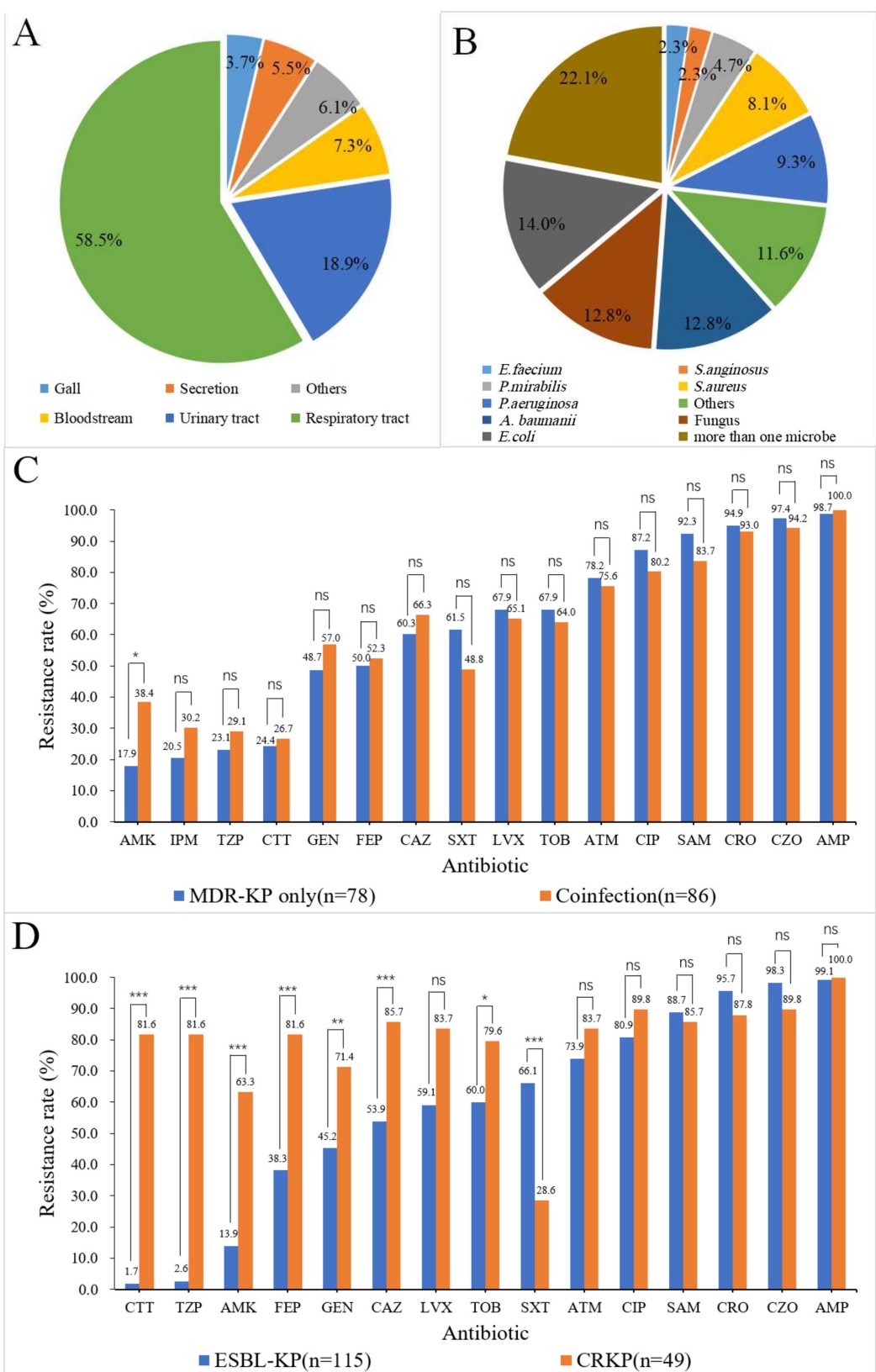

FIG 2 Microbiological characteristics of the 164 MDR-KP isolates included. (A) The isolation site of the 164 MDR-KP strains. (B) Distribution of microbes in the coinfection group. (C) Percentage of antimicrobial resistance of isolates in MDR-KP only and coinfection group. (D) Percentage of antimicrobial resistance of isolates in ESBL-KP and CRKP group. *P*-values were obtained (Continued on next page)

**Fig 2 (Continued)**

through χ based on the data. "ns" indicated no statistical difference between the two groups. "*" represented that the *P*-value for the comparison of two groups is less than 0.05. "**" represented that the *P*-value for the comparison of two groups is less than 0.01. "***" represented that the *P*-value for the comparison of two groups is less than 0.001. Abbreviations: *A. baumanii, Acinetobacter baumannii*; *E. coli, Escherichia coli*; *E. faecium, Enterococcus faecium*; *P. aeruginosa, Pseudomonas aeruginosa*; *P. mirabilis, Proteus mirabilis*; *S. aureus, Staphylococcus aureus*; *S. anginosus, Streptococcus anginosus*; MDR-KP, multidrug-resistant *Klebsiella pneumoniae*; ESBL-KP, extended-spectrum beta-lactamase-producing *Klebsiella pneumoniae*; CRKP, carbapenem-resistant *Klebsiella pneumoniae*; AMK, amikacin; IPM, imipenem; TZP, piperacillin/tazobactam; CTT, cefotetan; GEN, gentamicin; FEP, cefepime; SXT, trimethoprim-sulfamethoxazole; CAZ, ceftazidime; LVX, levofloxacin; TOB, tobramycin; ATM, aztreonam; CIP, ciprofloxacin; SAM, ampicillin/sulbactam; CRO, ceftriaxone; CZO, cefazolin; AMP, ampicillin.

combined antibiotics were similar in the ESBL-KP and CRKP groups (Fig. 4). The use of immunosuppressive (57.0% vs 48.7%, *P* = 0.290) and hormonal therapy (68.6% vs 64.1%, *P* = 0.542) was more common in the coinfected patients, as well. In addition, the majority

**TABLE 3** Drug treatment and clinical outcomes in MDR-KP only and coinfection groups[b]

| Variable | Total (*n* = 164) | MDR-KP only (*n* = 78) | Coinfection (*n* = 86) | *P*-value[a] |
|---|---|---|---|---|
| Treatment with antibiotics | | | | |
| Empirical antibiotic therapy | 139 (84.8%) | 66 (84.6%) | 73 (84.9%) | 0.962 |
| Second/third-generation cephalosporins | 84 (51.2%) | 36 (46.2%) | 48 (55.8%) | 0.216 |
| Piperacillin–tazobactam | 43 (26.2%) | 20 (25.6%) | 23 (26.7%) | 0.873 |
| Carbapenems | 20 (12.2%) | 11 (14.1%) | 9 (10.5%) | 0.477 |
| Quinolones | 32 (19.5%) | 14 (17.9%) | 18 (20.9%) | 0.630 |
| Fosfomycin | 7 (4.3%) | 5 (6.4%) | 2 (2.3%) | 0.191 |
| Antifungal agent | 15 (9.1%) | 4 (5.1%) | 11 (12.8%) | 0.089 |
| Single DAT | | | | |
| β-Lactams | 46 (28%) | 26 (33.3%) | 20 (23.3%) | 0.151 |
| Quinolones | 7 (4.3%) | 4 (5.1%) | 3 (3.5%) | 0.604 |
| Others | 7 (4.3%) | 3 (3.8%) | 4 (4.7%) | 0.799 |
| Combined DAT | | | | |
| Combination of two antimicrobials | 31 (18.9%) | 9 (11.5%) | 22 (25.6%) | **0.022** |
| Combination of ≥ triple antimicrobials | 31 (18.9%) | 8 (10.3%) | 23 (26.7%) | **0.007** |
| Concomitant drugs | | | | |
| Acid-suppressing agent | 88 (53.7%) | 35 (44.9%) | 53 (61.6%) | **0.032** |
| Immunosuppressant | 87 (53%) | 38 (48.7%) | 49 (57.0%) | 0.290 |
| Hormone | 109 (66.5%) | 50 (64.1%) | 59 (68.6%) | 0.542 |
| Clinical outcomes | | | | |
| LOS in ICU | 0 (0, 10.3) | 0 (0, 6.8) | 6.0 (0, 16.5) | **0.001** |
| Direct admission to ICU | 50 (30.5%) | 19 (24.4%) | 31 (36.0%) | 0.104 |
| LOS | 14.0 (8.0, 24.3) | 10.5 (6.3, 17.8) | 19.0 (9.0, 33.8) | **<0.001** |
| LOS after infection | 8.0 (4.0, 15.0) | 6.0 (4.0, 11.0) | 11.0 (5.3, 17.8) | **0.001** |
| Clinical stability | 62 (37.8%) | 30 (38.5%) | 32 (37.2%) | 0.869 |
| On the mend | 17 (10.4%) | 12 (15.4%) | 5 (5.8%) | **0.045** |
| Deterioration | 12 (7.3%) | 2 (2.6%) | 10 (11.6%) | **0.026** |
| Automatic discharge or transfer | 27 (16.5%) | 15 (19.2%) | 12 (14.0%) | 0.363 |
| Readmission | 18 (11.0%) | 7 (9.0%) | 11 (12.8%) | 0.435 |
| Reinfection | 7 (4.3%) | 4 (5.1%) | 3 (3.5%) | 0.604 |
| In-hospital mortality | 21 (12.8%) | 9 (11.5%) | 12 (14.0%) | 0.644 |
| 7-day mortality | 17 (10.4%) | 10 (12.8%) | 7 (8.1%) | 0.326 |
| 30-day mortality | 31 (18.9%) | 12 (15.4%) | 19 (22.1%) | 0.273 |
| 90-day mortality | 40 (24.4%) | 15 (19.2%) | 25 (29.1%) | 0.143 |

[a]*P*-values were obtained using χ for categorical variables and Kruskal Wallis test for continuous variables.
[b]Data are represented as no. (%) and IQR, interquartile range. Bold values indicate statistical significance (*P* < 0.05). MDR-KP, multidrug-resistant *Klebsiella pneumoniae*; DAT, definite antibiotic therapy; ICU, intensive care unit; LOS, length of hospital stay.

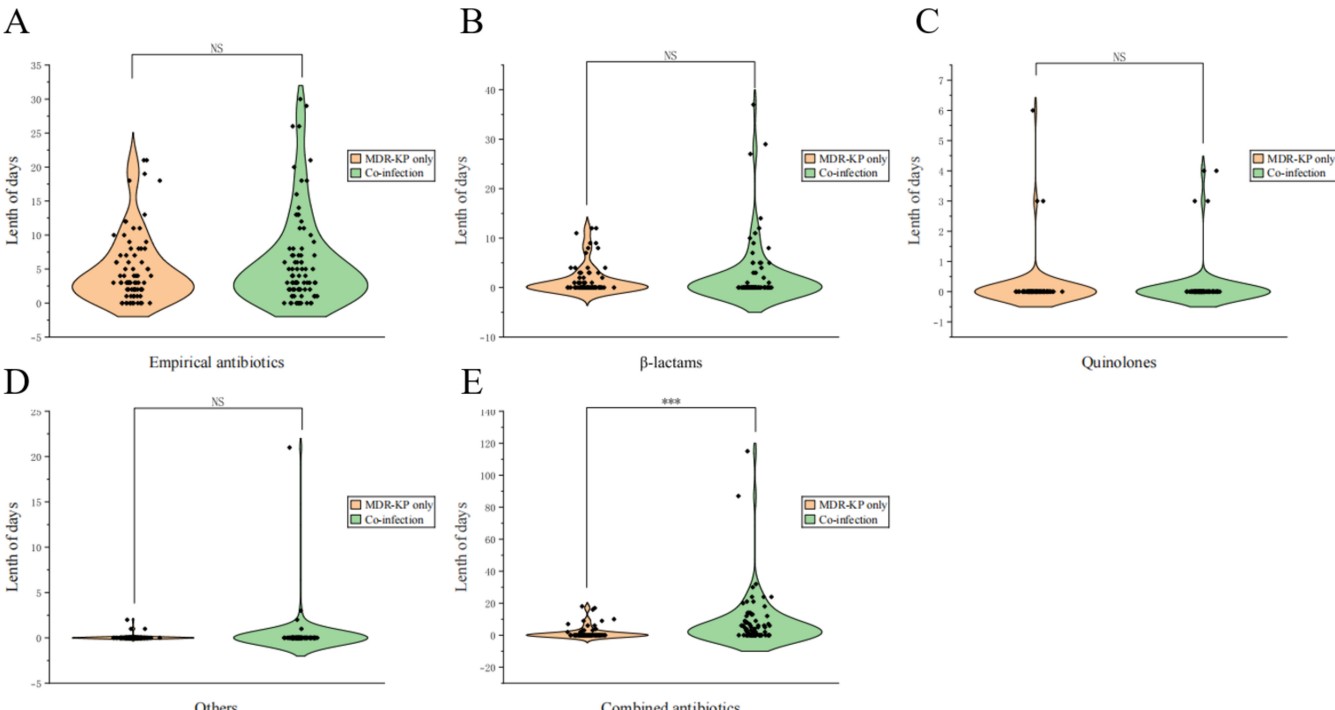

**FIG 3** Violin plot of duration of antibiotic therapy in the MDR-KP only and coinfection groups. (A) Empirical antibiotics, (B) β-lactams, (C) quinolones, (D) others, and (E) combined antibiotics for patient infection with MDR-KP. *P*-values were obtained through χ based on the data. "NS" indicated no statistical difference between the two groups. "***" represented that the *P*-value for the comparison of two groups is less than 0.001. Abbreviation: MDR-KP, multidrug-resistant *Klebsiella pneumoniae*.

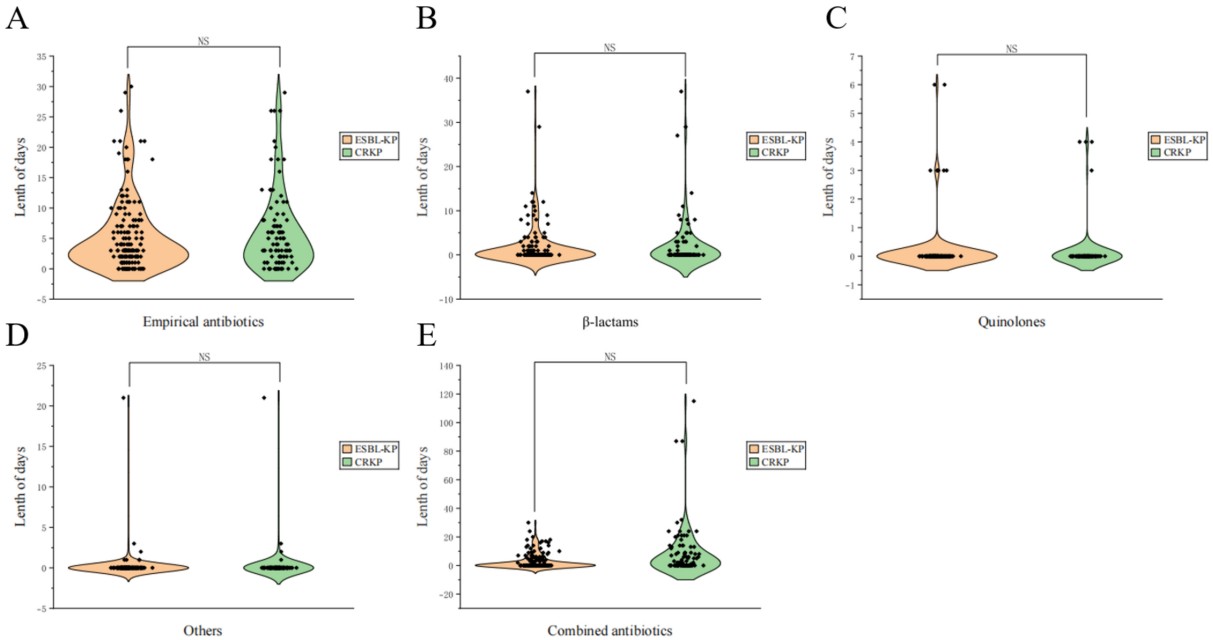

**FIG 4** Violin plot of duration of antibiotic therapy in the ESBL-KP and CRKP groups. (A) Empirical antibiotics, (B) β-lactams, (C) quinolones, (D) others, and (E) combined antibiotics for patient infection with MDR-KP. *P*-values were obtained through χ based on the data. "NS" indicated no statistical difference between the two groups. Abbreviations: ESBL, extended-spectrum beta-lactamase-producing *Klebsiella pneumoniae*; CRKP, carbapenem-resistant *Klebsiella pneumoniae*.

of patients were treated with acid-suppressing agents (including proton pump inhibitors and histamine-2 receptor antagonists) (61.6% vs 44.9%, *P* = 0.032) (Table 3).

Patients with coinfection tended to stay longer in the ICU than those with MDR-KP only (median, 6.0 vs 0 days, $P = 0.001$), and 1/3 of patients in the coinfection group were admitted to the ICU directly (36.0% vs 24.4%, $P = 0.104$). In addition, both the total length of hospital stay and post-infection hospital stay were significantly longer in the coinfection group (total: median, 19.0 vs 10.5 days, $P < 0.001$; post-infection: median, 11.0 vs 6.0 days, $P = 0.001$). Additionally, patients with coinfection tended to deteriorate (11.6% vs 2.6%, $P = 0.026$) and had a higher in-hospital mortality rate (14.0% vs 11.5%, $P = 0.644$) compared with the MDR-KP only group (Table 3). Compared with the ESBL-KP group, the CRKP group tended to have worse clinical outcomes (14.3% vs 4.3%, $P = 0.033$) and a significantly longer stay in the ICU (median, 7.0 vs 0 days, $P < 0.001$), with 51.0% of patients in the CRKP group admitted to the ICU directly (51.0% vs 21.7%, $P < 0.001$). In contrast, the probability of readmission (14.8% vs 2.0%, $P = 0.017$) and reinfection (6.1% vs 0.0%, $P = 0.024$) was higher in the ESBL-KP group (Table 4). Additionally, we analyzed 30-day all-cause mortality and plotted K-M curves, which showed that the 30-day all-cause mortality ($P = 0.03$) was lower in the coinfected group than that in the MDR-KP only group (Fig. 5A). Survival probability in the CRKP group was consistently

**TABLE 4** Drug treatment and clinical outcomes in ESBL-KP and CRKP groups[b]

| Variable | ESBL-KP ($n = 115$) | CRKP ($n = 49$) | $P$-value[a] |
|---|---|---|---|
| Treatment with antibiotics | | | |
| Empirical antibiotic therapy | 99 (86.1%) | 40 (81.6%) | 0.468 |
| Second/third-generation cephalosporins | 64 (55.7%) | 20 (40.8%) | 0.082 |
| Piperacillin–tazobactam | 27 (23.5%) | 16 (32.7%) | 0.221 |
| Carbapenems | 12 (10.4%) | 8 (16.3%) | 0.291 |
| Quinolones | 24 (20.9%) | 8 (16.3%) | 0.502 |
| Antifungal agent | 11 (9.6%) | 4 (8.2%) | 0.773 |
| Single DAT | | | |
| β-Lactams | 35 (30.4%) | 11 (22.4%) | 0.297 |
| Quinolones | 6 (5.2%) | 1 (2.0%) | 0.325 |
| Others | 5 (4.3%) | 2 (4.1%) | 0.938 |
| Combined DAT | | | |
| Combination of two antimicrobials | 21 (18.3%) | 10 (20.4%) | 0.748 |
| Combination of ≥ triple antimicrobials | 20 (17.4%) | 11 (22.4%) | 0.449 |
| Concomitant drugs | | | |
| Acid-suppressing agent | 63 (54.8%) | 25 (51.0%) | 0.658 |
| Immunosuppressant | 61 (53.0%) | 26 (53.1%) | 0.998 |
| Hormone | 74 (64.3%) | 35 (71.4%) | 0.379 |
| Clinical outcomes | | | |
| LOS in ICU | 0 (0, 8.0) | 7.0 (0, 18.0) | **<0.001** |
| Direct admission to ICU | 25 (21.7%) | 25 (51.0%) | **<0.001** |
| LOS | 13.0 (8.0, 22.5) | 15.0 (8.0, 33.0) | 0.226 |
| LOS after infection | 8.0 (4.0, 14.0) | 9.0 (3.0, 16.0) | 0.991 |
| Clinical stability | 43 (37.4%) | 19 (38.8%) | 0.867 |
| On the mend | 13 (11.3%) | 4 (8.2%) | 0.546 |
| Deterioration | 5 (4.3%) | 7 (14.3%) | **0.033** |
| Automatic discharge or transfer | 18 (15.7%) | 9 (18.4%) | 0.668 |
| Readmission | 17 (14.8%) | 1 (2.0%) | **0.017** |
| Reinfection | 7 (6.1%) | 0 (0.0%) | **0.024** |
| In-hospital mortality | 13 (11.3%) | 8 (16.3%) | 0.229 |
| 7-day mortality | 9 (7.8%) | 8 (16.3%) | 0.133 |
| 30-day mortality | 18 (15.7%) | 13 (26.5%) | 0.151 |
| 90-day mortality | 25 (21.7%) | 15 (30.6%) | 0.266 |

[a]$P$-values were obtained using χ for categorical variables and Kruskal Wallis test for continuous variables.
[b]Data are represented as no. (%) and IQR, interquartile range. Bold values indicate statistical significance ($P < 0.05$). ESBL, extended-spectrum beta-lactamase-producing *Klebsiella pneumoniae*; CRKP, carbapenem-resistant *Klebsiella pneumoniae*; DAT, definite antibiotic therapy; ICU, intensive care unit; LOS, length of hospital stay.

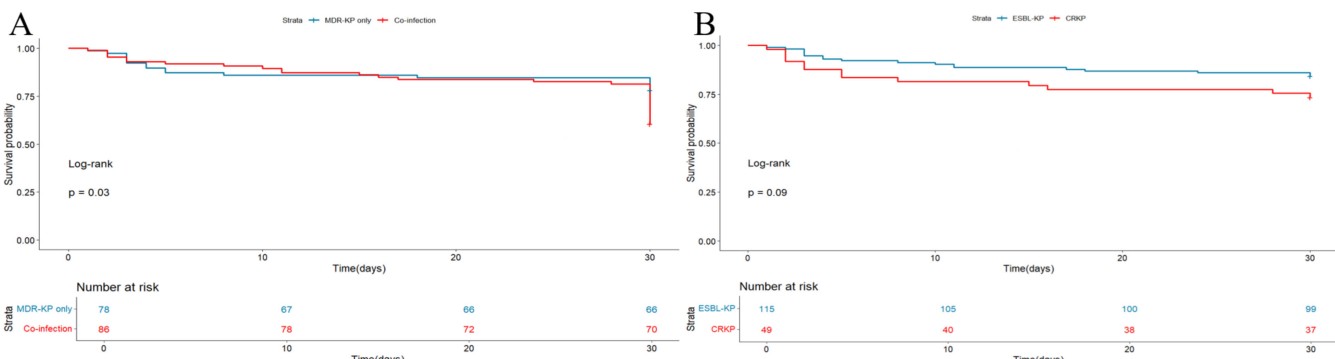

FIG 5 Kaplan-Meier survival analysis of different groups. (A) The 30-day mortality in the MDR-KP only and the coinfection groups. (B) The 30-day mortality in the ESBL-KP and the CRKP groups. The log-rank test was employed to assess statistical significance, and the corresponding $P$ values are provided. Abbreviations: MDR-KP, multidrug-resistant *Klebsiella pneumoniae*; ESBL, extended-spectrum beta-lactamase-producing *Klebsiella pneumoniae*; CRKP, carbapenem-resistant *Klebsiella pneumoniae*.

lower than that in the ESBL-KP group (Fig. 5B), although not statistically different ($P$ = 0.09).

## Logistic regression analysis

The relevant variables were included in logistic regression analysis to investigate the risk factors for coinfections (Fig. 6A). Multivariate logistic regression analysis showed that the nasogastric catheter (OR, 5.531; 95% CI, 1.437–19.926, $P$ = 0.012), as well as the venous and arterial catheter (OR, 5.182; 95% CI, 1.272–21.113, $P$ = 0.022), were independent risk factors for coinfection after adjustment for age and sex (Fig. 6A). The analysis also showed that tracheostomy (OR, 4.673; 95% CI, 1.153–18.937, $P$ = 0.031) and fiberoptic bronchoscopy (OR, 4.041; 95% CI, 1.305–12.516, $P$ = 0.015) were independent risk factors for CRKP infections after adjustment for age and sex (Fig. 6B).

## Predictive model for MDR-KP by logistic regression model

To predict the probability of coinfection, a nomogram with the independent risk factors was developed (Fig. 7A). Figure 7B demonstrates how the nomogram can be used to predict the risk of coinfection. After nasogastric catheter placement, patients infected with MDR-KP only had a risk score of 100 (probability = 0.738) for developing a coinfection. The discrimination of the prediction model was assessed by ROC curve, yielding an AUC of 0.773 (95% CI: 0.7054–0.8405) (Fig. 7C). The calibration curve demonstrated a good consistency between the observed and predicted values (Fig. 7D). Additionally, the nomogram was developed for the risk of CRKP infection (Fig. 8A). Following fiberoptic bronchoscopy, patients infected with ESBL-KP showed a risk score of 93.9 (probability = 0.467) for developing CRKP (Fig. 8B). Model discrimination was assessed by the ROC curve, giving an AUC of 0.752 (95% CI: 0.6739–0.8306) (Fig. 8C). The calibration curve indicated a good consistency between the observed and predicted values, as well (Fig. 8D).

## DISCUSSION

Several studies have highlighted the emergence of polymicrobial infections in patients with bloodstream infection or diabetic foot in the last decade (10, 11). However, our study involved patients with various conditions, encompassing a diversity of sources of infection. In clinical practice, early identification of patients coinfected with MDR-KP, as well as differentiation of CRKP from ESBL-KP infection, is vital to boost their survival prospects. In this study, we successfully constructed nomogram models to predict the probability of MDR-KP coinfection, as well as the occurrence of CRKP infection, through a retrospective cohort study. This model can enable early-stage screening of infections for

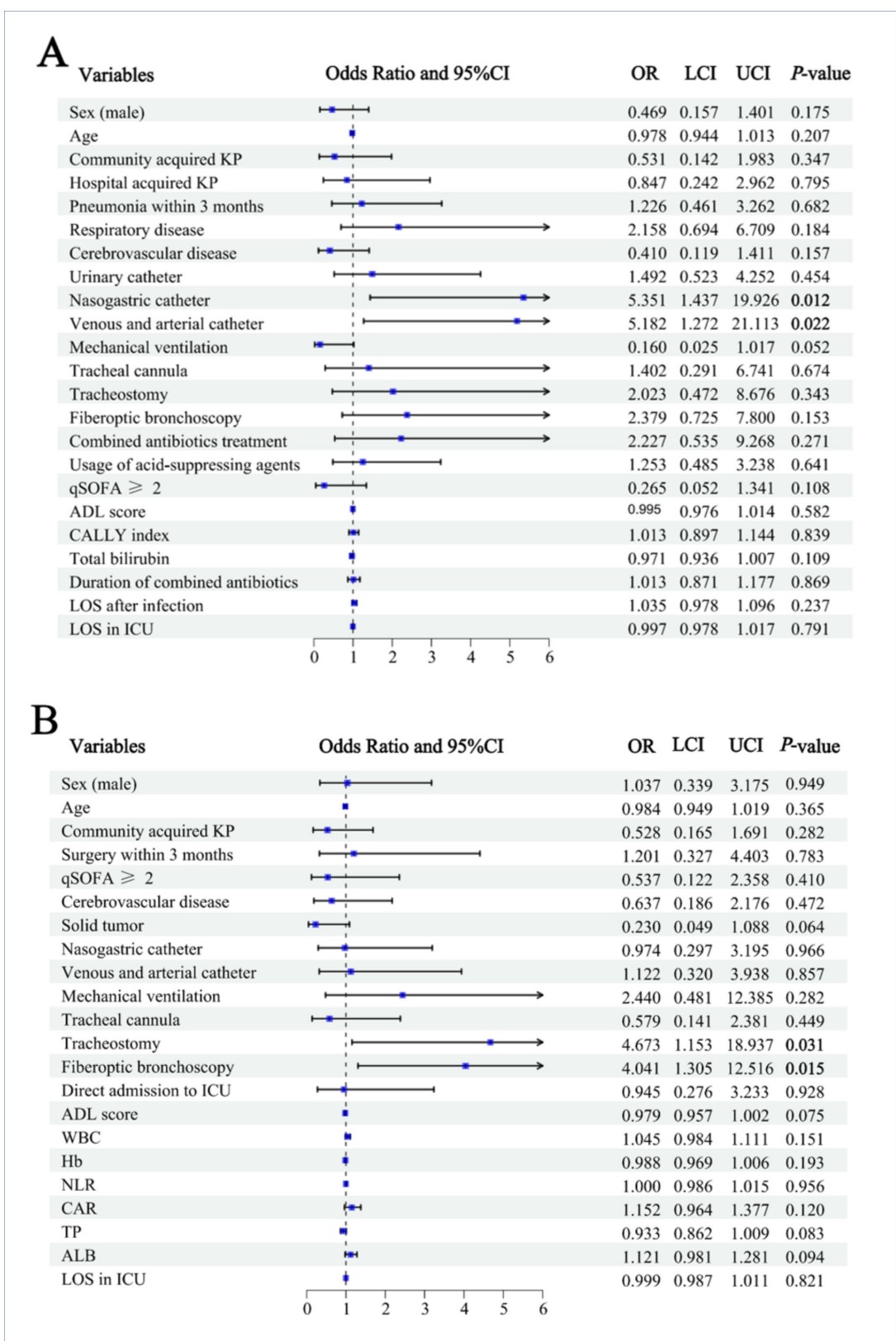

**FIG 6** Multivariate logistic analyses of risk factors. (A) Risk factors associated with the coinfection group (*n* = 86). (B) Risk factors associated with the CRKP group (*n* = 49). Numbers on the horizontal axis represented odds ratio (OR), which were used to assess the strength of association between exposure factors and outcomes; a higher value reflected a greater risk of the outcome linked to the exposure factor. Abbreviations: KP, *Klebsiella pneumoniae*; WBC, white blood cells; Hb, hemoglobin; CALLY index, C-reactive protein-albumin-lymphocyte index; NLR, neutrophil-to-lymphocyte ratio; CAR, C-reactive protein-to-albumin ratio; TP, total protein; ALB, albumin; LOS, length of stay; ICU, intensive care unit.

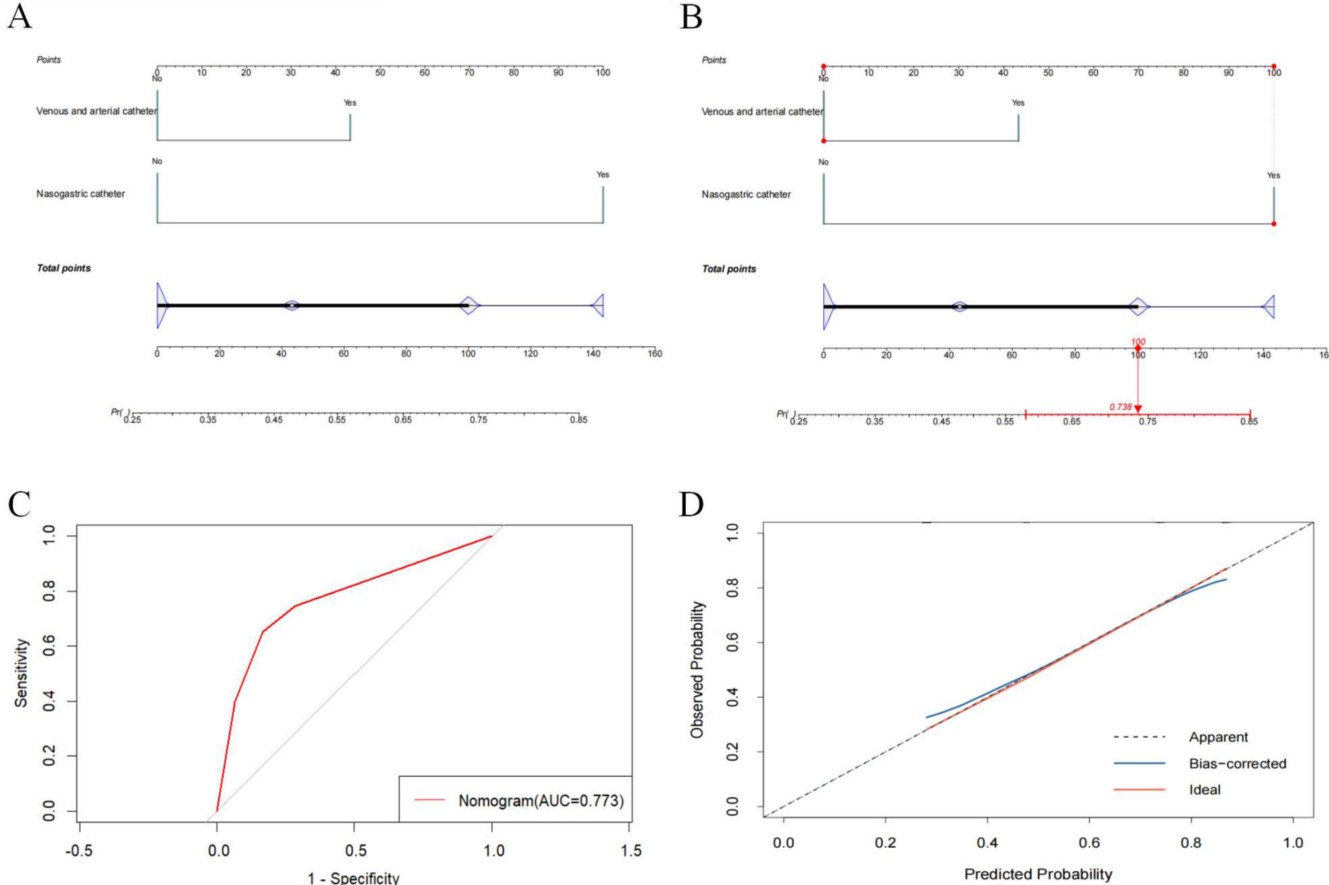

**FIG 7** Risk prediction model based on logistic regression results and its effectiveness evaluation. (A) Risk model of MDR-KP for coinfection. (B) Example of an application of the nomogram to predict the risk of coinfection. (C) Receiver-operating characteristic curve for predicting coinfection risk. AUC, area under the curve. (D) Calibration curve of the line graph model for predicting the risk of coinfection.

healthcare professionals. In contrast to traditional logistic regression analyses, our linear model calculated the scores corresponding to each influencing factor and matched the total scores to the predicted *P*-value, thus providing accurate predictions via simple graphical representations. The model has been widely applied to clinical cohort studies (20, 21). This may be the first reported predictive model for MDR-KP coinfections and CRKP infections, providing a reliable basis for reducing and preventing MDR-KP coinfections and CRKP infections.

ICUs have facilitated the creation, amplification, and spread of antibiotic resistance (22). In our study, 36.0% of coinfected patients and 51.0% of CRKP patients were admitted to the ICU directly, which was consistent with previous studies (13, 23). We found that nasogastric catheters, as well as venous and arterial catheters, were risk factors for coinfection after adjustment for age and sex, while tracheotomy and fiberoptic bronchoscopy were risk factors for CRKP infection. The nasopharynx, a mandatory route for bronchoscopy, is one of the most common colonization sites for KP (24). Both nasogastric tube intubation and fiberoptic bronchoscopy can potentially cause damage to the mucosal barrier of the nasopharynx (25, 26). Therefore, intubation or tracheotomy poses a high risk of damaging normal tissue, facilitating the invasion and adhesion of opportunistic pathogens to respiratory mucosa, which may subsequently be encapsulated by the biofilm. This nasopharyngeal or respiratory mucosal damage is particularly prevalent among critical patients who have coinfections or suffer from CRKP. As a result, invasive procedures can be seen as significant risk factors, which is consistent with many studies (16, 25, 26). However, prior antibiotic exposure was not observed in

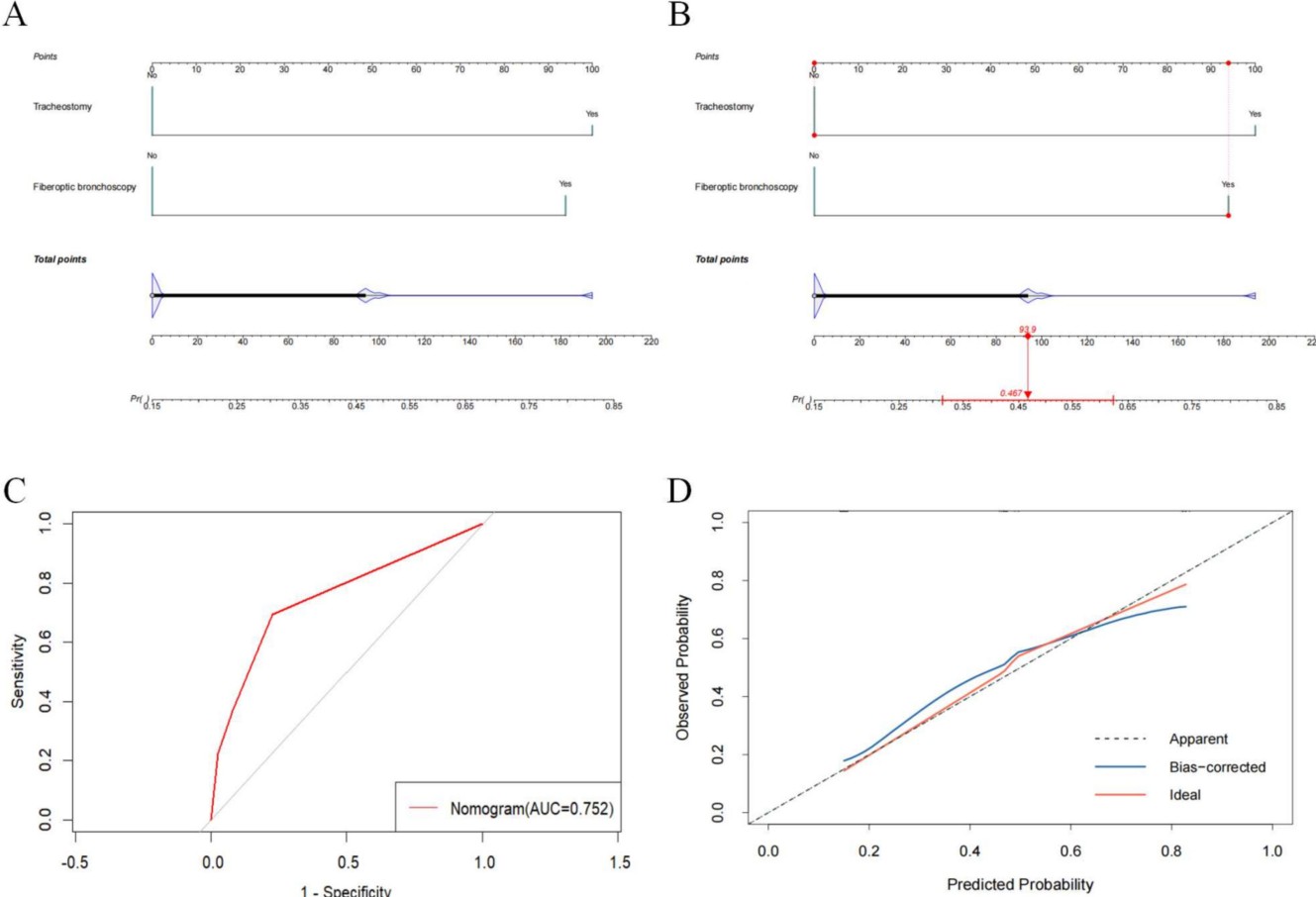

**FIG 8** Risk prediction model based on logistic regression results and its effectiveness evaluation. (A) Risk model of carbapenem-resistant *Klebsiella pneumoniae* (CRKP) infection. (B) Example of an application of the nomogram to predict the risk of CRKP infection. (C) Receiver-operating characteristic curve for predicting CRKP risk. AUC, area under the curve. (D) Calibration curve of the line graph model for predicting the risk of CRKP infection.

our study due to limited access to data. Simple cleaning and high-level disinfection on bronchoscopes may not always be adequately effective for eliminating the risk of infections caused by carbapenem-resistant Enterobacteriaceae (CRE) and related multidrug-resistant organisms (MDROs) (27). It is also essential to deepen healthcare workers' understanding of infections associated with invasive procedures and to strictly control the indications for the use of various invasive devices and catheters. Additionally, patients with coinfection were significantly more likely to develop pneumonia in the past 3 months, which may increase the risk of respiratory infections. Similarly, a study in India showed that MDR-KP is frequently isolated from sputum samples. However, studies conducted in eastern Saudi Arabia and Thailand found that the urethra was a common source and the most important colonization site for multidrug-resistant Gram-negative bacteria (23, 28). This might be attributed to differences in specific geographical regions, antibiotic use, healthcare settings, and patterns of drug resistance. Thus, in order to select empirical therapies that optimize patient outcomes, it is vital to understand the susceptibility patterns and distribution of drug-resistant bacteria in a given hospital setting.

The 30-day all-cause mortality rates in the coinfected group and the CRKP group were 22.1% and 26.5% respectively, in line with the range (16.9–36.1%) provided by the previous study (16, 29, 30). We assessed the severity of patients with MDR infection in terms of comorbidities, laboratory markers, and so on. The high mortality rate observed in this study can be attributed to several factors. (i) A large proportion of patients were

admitted to the ICU with multiple comorbidities (152/164, 92.7%), resulting in an overall higher mortality rate. Patients with coinfection were more likely to develop respiratory and cerebrovascular diseases, and the CRKP group was more likely to develop cerebrovascular diseases. Several studies have shown that comorbidities, especially chronic respiratory and cardiovascular diseases, were closely related to the poor outcomes during hospitalization (31, 32). (ii) Multidrug-resistant strains were inherently more broadly resistant to antibiotics. Resistance to three or more classes of antibiotics severely limited the therapeutic options. (iii) The CALLY index was lower in the coinfected group. As previously reported, the early decline in the CALLY index indicated a poor prognosis in patients (33). Moreover, a higher level of inflammation and a poorer nutritional status in the CRKP group signaled a poor prognosis (34). To our knowledge, there was no simple linear superposition between possible risk factors and elevating the risk of death. Additionally, there was no significant difference in 30-day mortality rates between the ESBL-KP and CRKP groups, likely due to their inherent criticality. Anyway, patients with coinfections and CRKP infections showed a significantly greater nursing and economic burden, prolonged ICU stays, and a higher probability of being exposed to various invasive procedures during hospitalization.

In our study, the resistance rate to aminoglycosides of the coinfection group was only 38.4%, which was between 29%–54.9% in a study conducted in Bucharest, Romania (35). Therefore, aminoglycosides can still be used to treat *Klebsiella pneumoniae* infections, particularly urinary tract infections. Even though the resistance to ciprofloxacin rose notably from 14.7% in 2017 to 26.5% in 2020 (36), the resistance rate to fluoroquinolone antibiotics in our study still far exceeded this rate. Compared with high levels of resistance to cephalosporins, quinolones, and aminoglycosides, the resistance rate of CRKP isolates to sulfamethoxazole-trimethoprim in this study was lower at 28.6%. This was consistent with the trend of sulfonamide antibiotic resistance in Bucharest, Romania, from 2019 to 2021, which dropped from 74% to 39.1% (35). Moreover, a study conducted in China from 2014 to 2022 showed that the resistance rate to sulfonamide antibiotics of *Klebsiella pneumoniae* isolates remained fairly stable at around 24% (37). In our study, we found that both the coinfected group and the CRKP group attempted to use two or more antibiotics to control infections, but the therapeutic effect seemed unsatisfactory. As indicated in this research, the key factor associated with favorable outcomes was not the number of drugs used but rather the type of antibiotics (at least two) that demonstrated activity against the pathogenic strains *in vitro* (38). We also observed the frequent use of acid suppressants, including proton pump inhibitors (PPIs) and histamine-2 receptor antagonists (H2-RAs), in patients with coinfection. Relevant studies have revealed that the use of such acid suppressants increased the risks of both community-acquired and hospital-acquired pneumonia (39, 40). On the one hand, inhibiting gastric acid secretion may promote the overproliferation of bacteria in the upper gastrointestinal tract, which then migrate to the respiratory tract through aspiration (40). On the other hand, PPIs may affect serum mucus secretion, providing favorable conditions for bacterial growth (41). Therefore, our study emphasizes careful assessment before prophylactic use of PPIs/H2-RAs in patients, as acid suppression therapy may increase the risk of nosocomial infections.

Several limitations of this study should be mentioned. First, although this study included more isolates from various sources compared with previous studies and covered nearly 3 years, its single-center design may limit the generalizability of the findings to the broader Chinese population, thus constraining its external validity. Future studies could improve the representativeness and reliability of the findings through a multi-center design with a larger sample size. Second, further detection of resistance genes in MDR-KP strains is required to investigate the resistance characteristics of MDR-KP and associated treatment methods. Finally, although we have attempted to control for confounding factors, such as gender and age, in this study, it is not possible to entirely eliminate the risk of selection bias. To enhance model accuracy, more clinical data will be gathered for optimization.

## Conclusion

In summary, coinfections and CRKP infections significantly increased morbidity and economic burden, leading to longer ICU stays and poorer prognoses. Coinfection may also lead to a higher 30-day mortality rate. In order to reverse the rising trend in mortality rate associated with coinfection and CRKP infection, certain measures need to be taken: (i) develop stricter protocols for terminal cleaning of rooms (especially ICUs), cleaning of equipment (such as bronchoscopes) and hand hygiene; (ii) conduct drug resistance gene testing in the healthcare environment and implement antimicrobial drug management plans to optimize antibiotic consumption and reduce the emergence and spread of multi-drug resistance.

### ACKNOWLEDGMENTS

This work was supported by the Major Project of Humanities and Social Sciences Research in Anhui (Grant no. SK2021ZD0032) and the Key Project of Natural Science Research of Higher Education Institutions in Anhui Province (Grant no. 2024AH050739).

The study was designed by W.X. and M.Z. Data collection was conducted by X.X.S., C.H.X., Z.Q.Z., C.C.Z., and C.Q. The analyses were performed by X.X.S., C.H.X., J.L., and Z.J.Z., X.X.S., C.H.X., Z.Q.Z., and X.L.K. drafted the manuscript. All authors have read and approved the final manuscript.

### AUTHOR AFFILIATIONS

[1]Department of Clinical Laboratory, The Affiliated Chaohu Hospital of Anhui Medical University, ChaoHu, Anhui, China
[2]Beijing Friendship Hospital, Capital Medical University, Beijing, China
[3]School of Basic Medical Sciences, Capital Medical University, Beijing, China
[4]Department of Blood Transfusion, The First Affiliated Hospital of Anhui Medical University, Hefei, Anhui, China

### AUTHOR ORCIDs

Xixi Song http://orcid.org/0009-0007-8984-0012
Chonghe Xu http://orcid.org/0009-0001-1377-6946
Zhongqi Zhu http://orcid.org/0009-0002-4408-3947
Wei Xu http://orcid.org/0000-0003-3874-4438
Mei Zhu http://orcid.org/0000-0003-3130-3672

### FUNDING

| Funder | Grant(s) | Author(s) |
| --- | --- | --- |
| Anhui Provincial Department of Education | 2024AH050739 | Mei Zhu |
| Anhui Provincial Department of Education | SK2021ZD0032 | Mei Zhu |

### AUTHOR CONTRIBUTIONS

Xixi Song, Data curation, Formal analysis, Methodology, Software, Writing – original draft | Chonghe Xu, Methodology, Software, Validation, Visualization, Writing – review and editing | Zhongqi Zhu, Methodology, Resources, Software, Visualization, Writing – review and editing | Chenchen Zhang, Data curation, Investigation, Visualization, Writing – review and editing | Chao Qin, Data curation, Formal analysis, Software | Juan Liu, Data curation, Resources, Writing – review and editing | Xiaoli Kong, Formal analysis, Validation, Visualization, Writing – review and editing | Zhijun Zhu, Data curation, Methodology, Validation | Wei Xu, Conceptualization, Formal analysis, Investigation, Project administration, Resources, Validation, Visualization, Writing – review and editing

| Mei Zhu, Conceptualization, Funding acquisition, Project administration, Resources, Supervision, Writing – review and editing

## DATA AVAILABILITY

The data sets generated and/or analyzed during the current study are available from the corresponding author upon reasonable request.

## ETHICS APPROVAL

Ethical approval was granted by the Institutional Review Board of the Affiliated Chaohu Hospital of Anhui Medical University (approval no. KYXM-202312-052). Patient privacy was protected by anonymizing and de-identifying the data set prior to statistical analysis. Given the retrospective nature of the study and adherence to the Declaration of Helsinki, the requirement for informed consent was waived by the Ethics Committee.

## ADDITIONAL FILES

The following material is available online.

### Open Peer Review

**PEER REVIEW HISTORY (review-history.pdf).** An accounting of the reviewer comments and feedback.

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
