## [Reviewer comments · mSystems]

Multidrug-resistant *Klebsiella pneumoniae* Coinfection with Multiple Microbes: A Retrospective Study on Its Risk Factors and Clinical Outcomes

Xixi Song, Chonghe Xu, Zhongqi Zhu, Chenchen Zhang, Chao Qin, Juan Liu, Xiaoli Kong, Zhijun Zhu, Wei Xu, and Mei Zhu

Corresponding Author(s): Xixi Song, Chaohu Hospital of Anhui Medical University

Review Timeline:

Submission Date:	December 23, 2024
Editorial Decision:	March 19, 2025
Revision Received:	April 14, 2025
Accepted:	May 29, 2025

Editor: Naama Geva-Zatorsky

Reviewer(s): Disclosure of reviewer identity is with reference to reviewer comments included in decision letter(s). The following individuals involved in review of your submission have agreed to reveal their identity: Yuanzhao Ding (Reviewer #1); Nana Ama Amisah (Reviewer #2); Mohamed Abdelmonem (Reviewer #3)

Transaction Report:

DOI: <https://doi.org/10.1128/msystems.01757-24>

Re: mSystems01757-24 (Multidrug-resistant *Klebsiella pneumoniae* Coinfection with Multiple Microbes: A Retrospective Study on Its Risk Factors and Clinical Outcomes)

Dear Dr. Mei Zhu:

Revision Guidelines

Sincerely,
Naama Geva-Zatorsky
Editor
mSystems

Reviewer #1 (Comments for the Author):

This is an interesting study that seems suitable for publication in mSystems. However, I believe the paper requires further revisions before it can be accepted. Below are my suggestions:

Ethical Approval: Human studies typically require approval from the Institutional Review Board (IRB) or the Institutional Animal

Care and Use Committee (IACUC). The authors should verify if this is applicable.

Microorganism Names (Lines 147 and 171): The names of microorganisms, such as *Pseudomonas aeruginosa* ATCC 27853 and *Escherichia coli* ATCC, as well as *Klebsiella pneumoniae*, should be italicized. Additionally, the full name should be provided at first mention, followed by abbreviations in subsequent mentions.

Table 1 Formatting: There are inconsistencies in capitalization, such as "Respiratory Disease" versus "Cerebrovascular disease." Please ensure consistency throughout the table.

Currency Specification in Table 1: If the authors are referring to United States Dollars, this should be explicitly stated, e.g., "United States Dollars" or "US Dollars," as many other countries also use "dollars" (e.g., Singapore Dollars).

Figure 5 Explanation: The numbers 0, 1, 2, 3, 4, 5, and 6 in the figure require clarification. The authors could briefly explain their meaning in the figure caption, such as whether larger numbers indicate higher or lower values, to assist readers in understanding the figure.

At its current stage, I cannot recommend this paper for publication. I suggest more additional reviews and revisions are needed to address these points comprehensively.

Reviewer #2 (Comments for the Author):

The manuscript presents a comprehensive investigation into the epidemiology, risk factors, clinical outcomes, and predictive modeling for multidrug-resistant *Klebsiella pneumoniae* (MDR-KP) and carbapenem-resistant *Klebsiella pneumoniae* (CRKP) infections at a tertiary hospital in China. The research is timely and important, given the increasing global prevalence of MDR organisms. The study employs robust statistical methods and clearly articulates the findings. However, there are several aspects that could be improved upon.

Major comment:

1-The methodology is generally well-described; however, more specific details regarding the patients' selection criteria for inclusion and exclusion would enhance the reproducibility of the study.

2- Clarify the statistical methods used for logistic regression analysis. For instance, provide more details on how the variables were selected for inclusion and the rationale behind this process.

Minor Comments:

Grammar and Language:

Review the manuscript for grammatical errors and sentence structure. For example, certain phrases can be simplified for clarity.

Ethical Considerations:

It is mentioned that the study was approved by an ethics board, but more context on obtaining patient consent and data handling in line with ethical practices

**Multidrug-resistant *Klebsiella pneumoniae* Coinfection with Multiple Microbes:**
**A Retrospective Study on Its Risk Factors and Clinical Outcomes**

Xixi Song^{1*}, Chonghe Xu^{2*}, Zhongqi Zhu^{1*}, Chenchen Zhang^{1*}, Chao Qin¹, Juan Liu¹,
Xiaoli Kong¹, Zhijun Zhu¹, Wei Xu^{3#} & Mei Zhu^{1#}

1. Department of Clinical Laboratory, The Affiliated Chaohu Hospital of Anhui Medical University,
ChaoHu 238000, Anhui, PR China

2. School of Basic Medical Sciences, Capital Medical University, Beijing 100069, PR China

3. Department of Blood Transfusion, The First Affiliated Hospital of Anhui Medical University,
Hefei 230022, Anhui, PR China

*These authors contributed equally to this work

**First author:**

**Mrs. Xixi Song**, Department of Clinical Laboratory, the Affiliated Chaohu Hospital of Anhui
Medical University, No. 64 Chaohu North Road, Chaohu 238000, Anhui, PR China. Email:
matexixi@163.com. ORCID: <https://orcid.org/0009-0007-8984-0012>.

**#Correspondence**

**Dr. Wei Xu**, Department of Blood Transfusion, the First Affiliated Hospital of Anhui Medical
University, 210 Jixi Road, Hefei 230022, Anhui, PR China. Email: weixu532@sina.com;
xuwei@ahmu.edu.cn. ORCID: <https://orcid.org/0000-0003-3874-4438>. Tel: +8655162922331.

**Dr. Mei Zhu**, Department of Clinical Laboratory, the Affiliated Chaohu Hospital of Anhui Medical
University, No. 64 Chaohu North Road, Chaohu 238000, Anhui, PR China.
Email: zhumei@ahmu.edu.cn; meizhu532@sina.com. ORCID: [https://orcid.org/0000-0003-3130-](https://orcid.org/0000-0003-3130-3672)
3672. Tel: +8655182324254.

**Abstract**

**Background:** The prevalence of multidrug-resistant *Klebsiella pneumoniae* (MDR-KP) is rising
globally. The aim of this study was to investigate the epidemiology, risk factors and clinical
outcomes of MDR-KP coinfections and infections with carbapenem-resistant *Klebsiella pneumoniae*
(CRKP) among patients in a tertiary hospital in China, and to establish an individualized linear
prediction model.

**Methods:** In this retrospective study, patients admitted between January 2021 and March
2024 with a diagnosis of MDR-KP infection were included. We recorded demographics,
comorbidities, laboratory indicators, therapeutic interventions, antibiotic susceptibility
results (AST) and analyzed clinical outcomes. Logistic regression models were employed to
evaluate the risk factors associated with MDR-KP coinfections and infections with CRKP.

**Results:** A total of 164 patients with MDR-KP infection were included. Of these patients, 78
(47.6%) were infected with MDR-KP only and 86 (52.4%) were coinfecting with other
microbes; 115 (70.1%) were infected with extended-spectrum beta-lactamase producing
*Klebsiella pneumoniae* (ESBL-KP), and 49 (29.9%) were infected with CRKP. The most
common source of infection in patients with MDR-KP infection was the respiratory tract
(96/164, 58.5%), followed by the urinary tract (31/164, 18.9%). Multivariate logistic
regression analysis showed that nasogastric catheters (OR 5.351, 95% CI 1.437-19.926, $P =$
0.012), as well as venous and arterial catheters (OR 5.182, 95% CI 1.272-21.113, $P = 0.022$)
were independent risk factors for coinfection. The total risk score for all factors was 143.3,
with a predicted risk rate ranging from 0.25 to 0.85. In the ROC curve analysis, the area under
the curve (AUC) for predicting coinfection using the total risk score was 0.773 (95% CI:
0.7054-0.8405). Tracheostomy (OR 4.673, 95% CI 1.153-18.937, $P = 0.031$) and fiberoptic
bronchoscopy (OR 4.041, 95% CI 1.305-12.516, $P = 0.015$) were independent risk factors for
infecting with CRKP, with a total risk score for all factors of 193.9, and a predicted risk rate
ranging from 0.15 ~ 0.85. In the ROC curve analysis, the area under the curve (AUC) for
predicting CRKP using the total risk score was 0.752 (95% CI: 0.6739-0.8306). Analysis on the
calibration curve indicated good agreement between the observed and predicted values. The
log-rank test was used to compare all-cause mortality between the two groups, and 30-day
mortality was higher in the coinfecting group than in the MDR-KP alone group ($P = 0.03$).

There was no significant difference in 30-day mortality between the CRKP group and
ESBL-KP group ($P = 0.09$).

**Conclusion:** This study successfully established a model based on risk factors, which has
good predictive value for both patients with coinfections and those with CRKP. Coinfections
and CRKP infections significantly increased overall mortality and economic burden, while
leading to poor prognosis in patients. These findings provided a basis for further clinical
research and refinement of strategies for managing MDR-KP coinfections and CRKP
infections.

**Keywords:** multidrug resistance, coinfection, *Klebsiella pneumoniae*, carbapenem resistance,
prediction model, mortality

1. Introduction

*Klebsiella pneumoniae*, a member of the Enterobacteriaceae family, is an opportunistic
pathogen that accounts for one-third of all gram-negative infections^[1]. It is included in the 2024
WHO Bacterial Priority Pathogens List (WHO BPPL) as well as ESKAPE pathogens^[2], and is
responsible for a wide range of infections, including urinary tract infections, pneumonia, bacteremia
and liver abscesses^[3]. Antibiotic resistance (AR) is a complex process involving multiple factors^[4].
For example, the selective pressure posed by the extensive use of antibiotics has facilitated the
emergence of multidrug-resistant *K. pneumoniae* (MDR-KP). Moreover, the conjugative transfer of
antibiotic resistance genes between bacterial species and genera has exacerbated the problem of
antibiotic resistance^[5]. A systematic review and meta-analysis showed that the prevalence of
nosocomial infections caused by MDR-KP and extended-spectrum beta-lactamase *K. pneumoniae*
(ESBL-KP) in south-eastern Asia were estimated to be 55% (95% CI, 9–96) and 27% (95% CI
32–100) respectively^[6]. In fact, multidrug-resistant bacteria are more likely to be found among
immunocompromised patients, featuring high mortality, reduced effectiveness of drugs, prolonged
hospitalization and high medical cost^[7]. Therefore, multidrug-resistant (MDR) *K. pneumoniae* is a
thorny issue for the global public health.

The incidence of polymicrobial infections, defined as infections that involve one or more types
of bacteria, viruses, fungi or parasites, has been rising for the past decades. Although polymicrobial
infections are increasingly diagnosed in critical patients, the major MDR infection control strategies

and antibiotic regimens are commonly based on the assumption of single-microbe infection^[8]. Poor
wound care, inappropriate use of antibiotics, poor hospital hygienic management could be the
common causes of polymicrobial infections^[9,10]. Polymicrobial infections are often associated with
poorer prognosis, including prolonged hospital and intensive care unit stays, increased incidence of
septic shock, increased need to amputate due to diabetic ulcers, and higher all-cause mortality^[11-13].

In an environment of rising multidrug-resistant infection rates, research on risk factors and
outcomes of multi-microbial infections is still limited and restricted to a single population of patients
with bloodstream infection or diabetic foot^[10,11]. Given that polymicrobial infections can be
identified in a variety of populations, this limitation may confound the proper perception of
polymicrobial infections among clinical healthcare professionals. And few studies have focused on
polymicrobial infections involving MDR-KP. Hence, we tried to elucidate the risk factors of
polymicrobial infections involving MDR-KP, to develop a predictive model, and to facilitate the
prevention and control of CRKP infection.

**2. Methods**

**2.1. Patients and grouping**

This study retrospectively included 164 patients infected with MDR-KP from a tertiary hospital
in China between January 2022 and March 2024. We only recorded the first admission if the patients
had been admitted more than once. As for the repeated positive microbiological cultures, we only
recorded the first positive result. In our study, infections with MDR-KP were divided into two groups
according to the category of infected microbes: the MDR-KP only and coinfecting with other
microbes. In addition, patients were also divided into two groups based on their resistance phenotype:
ESBL-KP group and CRKP group. This study has passed the ethical review of hospital's Institutional
Review Board (KYXM-202312-052).

**2.2 Clinical data**

Relevant clinical variables were gathered from the hospital's electronic medical record. The data
included age, sex, admission and discharge diagnoses, comorbidities (hypertension, diabetes mellitus,
respiratory disease, digestive system disease, cerebrovascular disease, cardiovascular disease,
chronic kidney disease, chronic liver disease and solid tumor), age-adjusted Charlson comorbidity
index (aCCI), quick sequential organ failure assessment (qSOFA), activity of daily living (ADL),

past history, invasive operations, therapeutic drugs, the source of infection, antibiotic susceptibility
results, laboratory test results at admission, admission to intensive care unit (ICU), the length of stay
(LOS) and clinical outcomes.

**2.3. Variable definitions**

MDR was defined as acquired resistance to at least one agent in three or more antimicrobial
drug categories^[14]. Coinfection referred to an infection in which MDR-KP and other microbes were
isolated from the same or different clinical specimen through the laboratory within seven days.
According to the 2018 guidelines from the Clinical and Laboratory Standards Institute (CLSI), a
strain is considered to exhibit carbapenem resistance if it has a minimum inhibitory concentration
(MIC) of meropenem or imipenem of 4 mg/L or higher, or an MIC of 2 mg/L or higher for
ertapenem^[15]. Patients were identified as immunocompetence or immunosuppression^[16]. qSOFA was
used for rapid screening of sepsis^[17]. Upon admission, the functional status of patients was evaluated
using the ADL index^[18], which measures a patient's capacity to carry out activities of daily living
independently. Patients' overall systemic health was assessed by aCCI^[19]. Administration of
antibiotics prior to antibiotic sensitivity test was defined as empirical treatment and otherwise as
definitive treatment. Combined antibiotic therapy included no less than two types of antibiotics and
lasted for more than 24 hours. The length of post-infection hospitalization was defined as the time
from the date of the first collection when the sample was tested positive to the date of discharge. In
this study, the 30-day and 90-day mortality were defined as all-cause deaths within 30 and 90 days
after infection.

**2.4. Microbiological analysis**

In our hospital, all isolates were identified by matrix-assisted laser desorption/ionization
time-of-flight mass spectrometry (MALDI-TOF-MS; Bruker Corporation, Karlsruhe, Germany).
According to the breakpoints for Enterobacteriaceae set by the Clinical and Laboratory Standards
Institute (CLSI) 2023 guidelines, antimicrobial susceptibility tests (AST) results were determined by
MIC values obtained by the dilution method and the diameters of the inhibition circle obtained by
the Kirby–Bauer's disk diffusion (KB) method. Sixteen antimicrobial agents were tested: amikacin,
imipenem, piperacillin/tazobactam, cefotetan, gentamicin, cefepime, trimethoprim-sulfamethoxazole,
ceftazidime, levofloxacin, tobramycin, aztreonam, ciprofloxacin, ampicillin/sulbactam, ceftriaxone,
cefazolin and ampicillin. Strains were considered sensitive or non-sensitive (either intermediate or

resistant) to each antibiotic tested. *Pseudomonas aeruginosa* ATCC 27853 and *Escherichia coli* ATCC
25922 were used as quality control strains in this experiment.

**2.5. Statistical analysis**

Variables were analyzed using IBM statistical product and service solutions (SPSS) (version
25.0). Categorical variables were expressed as frequencies and percentages, while continuous
variables were expressed as the mean \pm standard deviation for normally distributed data or as the
interquartile range (IQR) for those not normally distributed. Continuous variables that met the
normal distribution were analyzed with a two-independent sample t-test, while continuous variables
that met the skewed distribution were analyzed with the Mann-Whitney *U* test. Categorical variables
were tested using Chi-square test or Fisher's exact test, as appropriate. Risk factors associated with
MDR-KP coinfection with multiple microbes and CRKP were determined based on univariate and
multivariate logistic regressions. The statistical tests were analyzed using two-tailed and a *P* value
less than 0.05 was considered statistically significant. We used R4.1.1 software to establish the
prediction model of the nomogram. The caret package was applied for internal validation using the
bootstrap method, and the RMS package was used to calculate the consistency index (C-index). The
discriminatory capacity of the predictive model was evaluated using the area under the receiver
operating characteristic curve (AUC), and the calibration power was assessed through the calibration
plot. The Kaplan-Meier method was used to evaluate 30-day survival, with the log-rank test used to
compare survival curves.

**3. Results**

**3.1 Demographic clinical characteristics of patients**

The analysis included 164 patients who were firstly hospitalized for infection with MDR-KP
between January 2022 and March 2024. Of these patients, 78 (47.6%) were infected with MDR-KP
only and 86 (52.4%) were coinfecting with other microbes. Among the 164 patients enrolled, 115
(70.1%) were infected with extended-spectrum beta-lactamase producing *Klebsiella pneumoniae* and
49 (29.9%) were infected with Carbapenem-resistant *Klebsiella pneumoniae* (Figure 1).

Detailed demographic information and the clinical characteristics of the patients are presented
in Table 1 and Table 2. Infection with MDR-KP occurred more frequently among men (129/164,
78.7%). The study population was predominantly elderly, with a high proportion of patients aged \geq

65 (112/164, 68.3%). Only 18.3% (n = 30/164) of the isolates were healthcare-associated, while 28.0%
(n = 46/164) were community-acquired, and 53.7% (n = 88/164) were hospital-acquired. Strains
from the MDR-KP only group were common in hospital-acquired (32/78, 41.0%) and
community-acquired (31/78, 39.7%) cases. The strains in the coinfection group were predominantly
hospital-acquired and this proportion was significantly larger than that in the MDR-KP only group
(65.1% vs. 41.0%, $P = 0.002$) (Table 1). Strains in the CRKP group were prevalent in
hospital-acquired cases (31/49, 63.3%). More strains in the ESBL-KP group were obtained in the
community compared to the CRKP group (Table 2). Coinfected patients featured a greater proportion
of smoking (12.8% vs. 10.3%, $P = 0.613$), alcohol consumption (11.6% vs. 6.4%, $P = 0.247$), prior
surgical history (19.8% vs. 10.3%, $P = 0.091$) and prior hospitalization history (48.8% vs. 39.7%, $P =$
0.242). Compared to MDR-KP only group, patients in coinfection group were more likely to have
pneumonia (50.0% vs. 34.6%, $P = 0.047$) within three months (Table 1). Compared to ESBL-KP
group, CRKP group had a higher proportion of prior surgical history (24.5% vs. 11.3%, $P = 0.032$)
within three months (Table 2). And their aCCI scores were higher (median 5 vs. 4, $P = 0.168$) in the
coinfection group, although not statistically significant. The patient's ability to perform activities of
daily living, as measured by the ADL score at admission, was significantly lower in the coinfection
group (median 20 vs. 47.5, $P = 0.010$) and in the CRKP group (median 10 vs. 45, $P < 0.001$) (Table 1
and 2). The qSOFA ≥ 2 implied the possibility of organ dysfunction. A higher proportion of patients
in the coinfection group on the day of admission had a qSOFA score ≥ 2 (18.6% vs. 9.0%, $P = 0.003$)
(Table 1), and a higher proportion of patients in the CRKP group had a qSOFA score ≥ 2 compared
with the ESBL-KP group (24.5% vs. 9.6%, $P = 0.012$) (Table 2).

Most patients (152/164, 92.7%) had comorbidities, with respiratory disease (107/164, 65.2%)
being the most commonly reported, followed by hypertension (77/164, 47.0%), cerebrovascular
disease (70/164, 42.7%) and cardiovascular disease (48/164, 29.3%). There was a significant
difference in respiratory disease and cerebrovascular disease (76.7% vs. 52.6% and 52.3% vs. 32.1%,
$P = 0.001$ and $P = 0.009$, respectively) between the coinfection group and MDR-KP only group
(Table 1). Cerebrovascular disease was more likely to occur in the CRKP group compared to the
ESBL-KP group (57.1% vs. 36.5%, $P = 0.015$). Additionally, Patients infected with ESBL-KP were
at greater risk of developing solid organ tumors (22.6% vs. 8.2%, $P = 0.029$) (Table 2).

To be noted, the vast majority of patients (146/164, 89.0%) had undergone invasive procedures

during their hospitalizations. Urinary catheter (75.6% vs. 47.4%, $P < 0.001$), nasogastric catheter
 (65.1% vs. 16.7%, $P < 0.001$), venous and arterial catheter (48.8% vs. 17.9%, $P < 0.001$), mechanical
 ventilation (45.3% vs. 21.8%, $P = 0.001$), tracheal cannula (51.2% vs. 19.2%, $P < 0.001$),
 tracheostomy (26.7% vs. 5.1%, $P < 0.001$) and fiberoptic bronchoscopy (41.9% vs. 14.1%, $P < 0.001$)
 were more likely to be used in the coinfection group, compared with the MDR-KP only group (Table
 1). The probability of receiving nasogastric catheter (61.2% vs. 33.9%, $P = 0.001$), venous and
 arterial catheter (53.1% vs. 26.1%, $P = 0.001$), mechanical ventilation (59.2% vs. 23.5%, $P < 0.001$),
 tracheal cannula (57.1% vs. 27.0%, $P < 0.001$), tracheostomy (36.7% vs. 7.8%, $P < 0.001$) and
 fiberoptic bronchoscopy (55.1% vs. 17.4%, $P < 0.001$) was greater in the CRKP group compared to
 the ESBL-KP group (Table 2). Patients in the coinfection and CRKP groups spent more in the
 hospital, indicating a heavier financial burden caused by the infection. The coinfection group showed
 a lower CALLY index compared to the MDR-KP only group (median 0.18 vs. 0.23, $P = 0.014$),
 indicating a higher death risk. The coinfection group also showed a deficiency in total bilirubin
 (median 12.0 vs. 16.5, $P = 0.003$) than the MDR-KP only group at the first laboratory examination
 (Table 1). Compared to the ESBL-KP group, the white blood cell count (median 10.17 vs. 8.31, $P =$
 0.015), as well as the neutrophil count (median 8.57 vs. 6.49, $P = 0.007$), were higher in the CRKP
 group. The neutrophil-to-lymphocyte ratio (NLR), an important indicator for assessing the
 inflammatory response, was higher in the CRKP group compared to the ESBL-KP group (median
 9.63 vs. 5.64, $P = 0.018$). The C-reactive protein to albumin ratio (CAR) is a new composite index
 reflecting the inflammatory and nutritional status. In our study, the CRKP group showed a higher
 level of CAR (median 1.22 vs. 0.33, $P = 0.013$) and a lower protein synthesis capacity (total protein
 and albumin) (Table 2), indicating a poor prognosis.

**Table 1.** Comparison of clinical characteristics in MDR-KP only and coinfection groups.

Variables	Total (n = 164)	MDR-KP only (n=78)	Coinfection (n=86)	P-value*
Sex (male)	129 (78.7%)	64 (82.1%)	65 (75.6%)	0.313
Age group, y				
≤35	7 (4.3%)	4 (5.1%)	3 (3.5%)	0.604
36–64	45 (27.4%)	20 (25.6%)	25 (29.1%)	0.623
≥ 65	112 (68.3%)	54 (69.2%)	58 (67.4%)	0.806
Infection type				
Community acquired	46 (28.0%)	31 (39.7%)	15 (17.4%)	0.001
Healthcare associated	30 (18.3%)	15 (19.2%)	15 (17.4%)	0.767

[revised manuscript text omitted]

Variables	ESBL-KP (n=115)	CRKP (n=49)	P -value*
Sex (male)	90 (78.3%)	39 (79.6%)	0.849
Age group, y			
≤35	4 (3.5%)	3 (6.1%)	0.457
36–64	34 (29.6%)	11 (22.4%)	0.350
≥ 65	77 (67.0%)	35 (71.4%)	0.573
Infection type			
Community acquired	40 (34.8%)	6 (12.2%)	0.003
Healthcare associated	18 (15.7%)	12 (24.5%)	0.180
Hospital acquired	57 (49.6%)	31 (63.3%)	0.107
Past history in 3 months			
Smoking	12 (10.4%)	7 (14.3%)	0.481

Alcohol consumption	12 (10.4%)	3 (6.1%)	0.364
Surgical history	13 (11.3%)	12 (24.5%)	0.032
Hospitalization history	49 (42.6%)	24 (49%)	0.452
Prior infection history within 3 months			
Pneumonia	46 (40.0%)	24 (49.0%)	0.287
Hepatitis	45 (39.1%)	21 (42.9%)	0.656
Urinary tract infection	21 (18.3%)	6 (12.2%)	0.342
Skin and soft tissue infection	6 (5.2%)	1 (2.0%)	0.325
Bloodstream infection	5 (4.3%)	3 (6.1%)	0.636
Fever $\geq 72^{\circ}\text{H}$	17 (14.8%)	12 (24.5%)	0.136
aCCI	4 (3, 6)	4 (3, 5)	0.191
ADL score at admission	45 (12.5, 77.5)	10 (0, 30)	<0.001
qSOFA ≥ 2 at admission	11 (9.6%)	12 (24.5%)	0.012
Comorbidities			
Hypertension	52 (45.2%)	25 (51.0%)	0.496
Diabetes mellitus	23 (20.0%)	12 (24.5%)	0.521
Respiratory Disease	71 (61.7%)	36 (73.5%)	0.149
Digestive system disease	25 (21.7%)	11 (22.4%)	0.920
Cerebrovascular disease	42 (36.5%)	28 (57.1%)	0.015
Cardiovascular disease	33 (28.7%)	15 (30.6%)	0.805
Chronic kidney disease	27 (23.5%)	14 (28.6%)	0.491
Chronic liver disease	22 (19.1%)	9 (18.4%)	0.909
Solid tumor	26 (22.6%)	4 (8.2%)	0.029
Immunosuppression	74 (64.3%)	31 (63.3%)	0.895
Invasive procedure			
Urinary catheter	66 (57.4%)	36 (73.5%)	0.052
Nasogastric catheter	39 (33.9%)	30 (61.2%)	0.001
T-tube intubation	5 (4.3%)	2 (4.1%)	0.938
Venous and arterial catheter	30 (26.1%)	26 (53.1%)	0.001
Mechanical ventilation	27 (23.5%)	29 (59.2%)	<0.001
Tracheal cannula	31 (27.0%)	28 (57.1%)	<0.001
Tracheostomy	9 (7.8%)	18 (36.7%)	<0.001
Endoscope	15 (13.0%)	4 (8.2%)	0.371
Fiberoptic bronchoscopy	20 (17.4%)	27 (55.1%)	<0.001
Others	24 (20.9%)	14 (28.6%)	0.285
Related cost (dollars)[#]			
Total cost	2622.1 (1267.0, 6870.2)	4492.0 (2556.0, 10759.7)	0.001
Medical fee	785.3 (284.8, 1830.9)	1306.4 (620.3, 3027.3)	0.004
Antibacterial agent cost	195.8 (88.0, 525.9)	258.0 (117.7, 829.8)	0.227
Laboratory findings at admission			
WBC, $\times 10^9/\text{L}$	8.31 (5.75, 11.65)	10.17 (7.15, 14.80)	0.015
Neu, $\times 10^9/\text{L}$	6.49 (3.96, 9.59)	8.57 (5.91, 12.99)	0.007
Lym, $\times 10^9/\text{L}$	1.05 (0.65, 1.55)	1.11 (0.54, 1.53)	0.956

Mon, $\times 10^9/L$	0.52 (0.34, 0.67)	0.49 (0.33, 0.88)	0.407
PLT, $\times 10^9/L$	185 \pm 79	197 \pm 94	0.376
Hb, g/L	120 (103, 133)	110 (85, 125)	0.010
NLR	5.64 (3.19, 12.00)	9.63 (5.16, 17.14)	0.018
PLR	174.73 (119.52, 251.25)	200.00 (107.69, 307.21)	0.052
CAR	0.33 (0, 1.95)	1.22 (0.10, 3.82)	0.013
CALLY index	0.30 (0.04, 1.74)	0.10 (0.02, 0.71)	0.416
ALT, U/L	25 (17, 44)	28 (15, 41)	0.793
AST, U/L	27 (20, 47)	34 (22, 47)	0.159
TB, $\mu\text{mol/L}$	14.0 (9.6, 20.2)	12.0 (9.0, 17.0)	0.224
TP, g/L	64.9 \pm 8.4	60.1 \pm 9.7	0.003
ALB, g/L	36.2 \pm 5.9	33.1 \pm 7.1	0.006
Cr, $\mu\text{mol/L}$	72.0 (56.5, 105.5)	76.0 (57.0, 107.0)	0.931
Cys C, mg/L	1.19 (0.93, 1.70)	1.14 (0.87, 1.69)	0.430
Bun, mmol/L	7.4 (4.9, 10.4)	7.8 (4.8, 12.2)	0.372

* *P*-values are obtained using chi-square for categorical variables and Kruskal Wallis test for continuous
 variables. # 1 dollar = 7.0978 Chinese Yuan as of March, 2024. Data are represented as No. (%), IQR (interquartile
 range) and mean \pm SEM (standard error of the mean). Bold values indicate statistical significance (*P* < 0.05).

Abbreviations: ESBL-KP, extended-spectrum beta-lactamase producing *Klebsiella pneumoniae*; CRKP,
 carbapenem-resistant *Klebsiella pneumoniae*; aCCI, age-adjusted Charlson comorbidity index; ADL, activity of
 daily living; qSOFA, quick sequential organ failure assessment; WBC, white blood cells; Neu, neutrophils; Lym,
 lymphocytes; Mon, monocytes; PLT, platelet counts; Hb, hemoglobin; NLR, neutrophil to lymphocyte ratio; PLR,
 platelet to lymphocyte ratio; CAR, C-reactive protein to albumin ratio; CALLY index, C-reactive
 protein-albumin-lymphocyte index; ALT, alanine aminotransferase; AST, aspartate aminotransferase; TB, total
 bilirubin; TP, total protein; ALB, albumin; Cr, creatinine; Cys C, cystatin C; Bun, blood urea nitrogen.

3.2 The microbiological characteristics of patients

The isolation sites of the MDR-KP strains were analysed (Figure 2a). The most common source
 was respiratory tract (96/164, 58.5%), followed by urinary tract (31/164, 18.9%) and bloodstream
 infection (12/164, 7.3%), etc. Among patients coinfecting with only one type of bacterium (Figure 2b),
 the largest proportion were infected with *Escherichia coli* (12/86, 14.0%), followed by *Acinetobacter*
 *baumannii* (11/86, 12.8%) and *Pseudomonas aeruginosa* (8/86, 9.3%). Coinfections with
 gram-positive bacteria have also been recorded, which included *Staphylococcus aureus* (7/86, 8.1%)
 and *Enterococcus faecium* (2/86, 2.3%). The coinfecting fungi were all *Candida* species (11/86, 12.8%)
 Furthermore, 19 patients (22.1%) were coinfecting with more than one microbe.

[revised manuscript text omitted]

* *P*-values are obtained using chi-square for categorical variables and Kruskal Wallis test for continuous
variables. Data are represented as No. (%) and IQR, interquartile range. Bold values indicate statistical significance
($P < 0.05$).

Abbreviations: ESBL, extended-spectrum beta-lactamase producing *Klebsiella pneumoniae*; CRKP,
carbapenem-resistant *Klebsiella pneumoniae*; DAT, definite antibiotic therapy; ICU, intensive care unit; LOS,
length of hospital stay.

**3.4 Logistic regression analysis**

Relevant variables were included in logistic regression analyses to investigate the risk factors
for coinfections (Figure 6a). Multivariate logistic regression analysis showed that the nasogastric
catheter (OR 5.531, 95% CI 1.437-19.926, $P = 0.012$) and the venous and arterial catheter (OR 5.182,
95% CI 1.272- 21.113, $P = 0.022$) were independent risk factors for coinfection after adjustment for
age and sex (Figure 6a). The analysis also showed that tracheostomy (OR 4.673, 95% CI
1.153-18.937, $P = 0.031$) and fiberoptic bronchoscopy (OR 4.041, 95% CI 1.305-12.516, $P = 0.015$)
were independent risk factors for CRKP infections after adjustment for age and sex (Figure 6b).

**3.5 Predictive model for MDR-KP by logistic regression model**

A line graph model was developed to assess the risk of coinfection using R software, with the
variables of multiple logistic regression analysis serving as the predictive factors (Figure 7a). For
instance, in the case of a patient infected solely with MDR-KP, the scores for non-venous and arterial
catheters and nasogastric catheters were 0 and 100, respectively, leading to a cumulative score of 100
(Figure 7b). Calibration analysis indicated that the line graph model developed in this study predicts
the risk of coinfection with a consistency index of 0.773 (95% CI: 0.7054–0.8405) (Figure 7c). The
calibration curve demonstrated a good consistency between the observed and predicted values

(Figure 7d). Additionally, a line graph model was developed for the risk of CRKP infection using R
software (Figure 8a). For patients infected with ESBL-KP, the scores of non-tracheostomy and
fiberoptic bronchoscopy were 0 and 93.9, respectively, leading to a total score of 93.9 (Figure 8b).
Calibration analysis indicated that the line graph model developed in this study predicts the risk of
CRKP infection with a consistency index of 0.752 (95% CI: 0.6739-0.8306) (Figure 8c). The
calibration curve indicated a good consistency between the observed and predicted values as well
(Figure 8d).

**4. Discussion**

[revised manuscript text omitted]

Several limitations of this study should be mentioned. Firstly, although this study included more
isolates from various sources compared to previous studies and covered nearly three years, its
single-center design may limit the generalizability of the findings to the broader Chinese population,
thus constraining its external validity. Future studies could improve the representativeness and
reliability of the findings through a multi-center design with a larger sample size. Secondly, further
detection of resistance genes in MDR-KP strains is required to investigate the resistance
characteristics of MDR-KP and associated treatment methods. Finally, although we have attempted
to control for confounding factors such as gender and age in this study, it is not possible to entirely
eliminate the risk of selection bias. To enhance model accuracy, more clinical data will be gathered
for optimization.

**5. Conclusion**

In summary, coinfections and CRKP infections significantly increased morbidity and economic
burden, leading to longer ICU stays, and poorer prognoses. Coinfection may also lead to a higher
30-day mortality rate. In order to reverse the rising trend in mortality rate associated with coinfection
and CRKP infection, certain measures need to be taken: (1) develop stricter protocols for terminal
cleaning of rooms (especially ICUs), cleaning of equipment (such as bronchoscopes) and hand
hygiene; (2) conduct drug resistance gene testing in the healthcare environment and implement
antimicrobial drug management plans to optimize antibiotic consumption and reduce the emergence
and spread of multi-drug resistance.

**Funding Declaration**

This work was supported from Major Project of Humanities and Social Sciences Research in
Anhui (grant no. SK2021ZD0032), Anhui Medical University Clinical and Early Discipline
Co-construction Project and Key Project of Natural Science Research of Higher Education
Institutions in Anhui Province (Grant no. 2024AH050739).

**Author contributions**

The study was designed by W Xu and M Zhu. Data collection was conducted by XX Song, CH

Xu, ZQ Zhu, CC Zhang and C Qin. The analyses were performed by XX Song, CH Xu, J Liu and ZJ
Zhu. XX Song, CH Xu, ZQ Zhu and XL Kong drafted the manuscript. All authors have read and
approved the final manuscript.

**Consent for publication**

All authors read and approved the final manuscript.

**Competing interests**

All authors declare that they have no competing interests.

**References**

- [1] Navon-Venezia S, Kondratyeva K, Carattoli A. Klebsiella pneumoniae: A major worldwide
source and shuttle for antibiotic resistance[J]. FEMS Microbiology Reviews, 2017, 41(3):
252-275.
- [2] Mouanga-Ndzime Y, Onanga R, Longo-Pendy N M, et al. Epidemiology of community origin
of major multidrug-resistant ESKAPE uropathogens in a paediatric population in south-east
gabon[J]. Antimicrobial Resistance & Infection Control, 2023, 12(1): 47.
- [3] Soliman E A. Exploring AMR and virulence in klebsiella pneumoniae isolated from humans
and pet animals: A complement of phenotype by WGS-derived profiles in a one health study in
egypt[J].One Health, 2024, 19: 100904.
- [4] Watkins R R, Bonomo R A. Overview: Global and local impact of antibiotic resistance[J].
Infectious Disease Clinics of North America, 2016, 30(2): 313-322.
- [5] Moglad E, Alanazi N, Altayb H N. Genomic study of chromosomally and plasmid-mediated
multidrug resistance and virulence determinants in klebsiella pneumoniae isolates obtained
from a tertiary hospital in al-kharj, KSA[J]. Antibiotics, 2022, 11(11): 1564.
- [6] Salawudeen A, Raji Y E, Jibo G G, et al. Epidemiology of multidrug-resistant klebsiella
pneumoniae infection in clinical setting in south-eastern asia: A systematic review and
meta-analysis[J]. Antimicrobial Resistance & Infection Control, 2023, 12(1): 142.
- [7] Tanwar J, Das S, Fatima Z, et al. Multidrug resistance: An emerging crisis[J]. Interdisciplinary
Perspectives on Infectious Diseases, 2014, 2014: 1-7.
- [8] Goodman K, Simner P, Tamma P, et al. Infection control implications of heterogeneous
resistance mechanisms in carbapenem-resistant enterobacteriaceae (CRE)[J]. Expert Review of
Anti-infective Therapy, 2016, 14(1): 95-108.
- [9] Qamar M U, Rizwan M, Uppal R, et al. Antimicrobial susceptibility and clinical characteristics
of multidrug-resistant polymicrobial infections in pakistan, a retrospective study 2019–2021[J].
Future Microbiology, 2023, 18(17): 1265-1277.
- [10] Kim H J, Na S W, Alodaini H A, et al. Prevalence of multidrug-resistant bacteria associated
with polymicrobial infections[J]. Journal of Infection and Public Health, 2021, 14(12):
1864-1869.
- [11] Zheng C, Cai J, Liu H, et al. Clinical characteristics and risk factors in mixed-enterococcal
bloodstream infections[J]. Infection and Drug Resistance, 2019, Volume 12: 3397-3407.

- [12] Shettigar K, Bhat D V, Satyamoorthy K, et al. Severity of drug resistance and co-existence of
enterococcus faecalis in diabetic foot ulcer infections[J]. *Folia Microbiologica*, 2018, 63(1):
115-122.
- [13] Tadese B K, DeSantis S M, Mgbere O, et al. Clinical outcomes associated with co-infection of
carbapenem-resistant enterobacterales and other multidrug-resistant organisms[J]. *Infection*
*Prevention in Practice*, 2022, 4(4): 100255.
- [14] Magiorakos A P, Srinivasan A, Carey R B, et al. Multidrug-resistant, extensively drug-resistant
and pandrug-resistant bacteria: An international expert proposal for interim standard definitions
for acquired resistance[J]. *Clinical Microbiology and Infection*, 2012, 18(3): 268-281.
- [15] Humphries R, Bobenchik A M, Hindler J A, et al. Overview of changes to the clinical and
laboratory standards institute *performance standards for antimicrobial susceptibility testing*,
M100, 31st edition[J]. *Journal of Clinical Microbiology*, 2021, 59(12): e00213-21.
- [16] Liu Y, Huang L, Cai J, et al. Clinical characteristics of respiratory tract infection caused by
klebsiella pneumoniae in immunocompromised patients: A retrospective cohort study[J].
*Frontiers in Cellular and Infection Microbiology*, 2023, 13: 1137664.
- [17] Evans L, Rhodes A, Alhazzani W, et al. Surviving sepsis campaign: International guidelines for
management of sepsis and septic shock 2021[J]. *Intensive Care Medicine*, 2021, 47(11):
1181-1247.
- [18] Zhou J, Xu Y, Yang D, et al. Risk prediction models for disability in older adults: A systematic
review and critical appraisal[J]. *BMC Geriatrics*, 2024, 24(1): 806.
- [19] Chen J, Allel K, Zhuo C, et al. Extended-spectrum β -lactamase-producing escherichia coli and
klebsiella pneumoniae: Risk factors and economic burden among patients with bloodstream
infections[J]. *Risk Management and Healthcare Policy*, 2024, Volume 17: 375-385.
- [20] Wu J, Zhang H, Li L, et al. A nomogram for predicting overall survival in patients with
low-grade endometrial stromal sarcoma: A population-based analysis[J]. *Cancer*
*Communications*, 2020, 40(7): 301-312.
- [21] Lv J, Liu Y Y, Jia Y T, et al. A nomogram model for predicting prognosis of obstructive
colorectal cancer[J]. *World Journal of Surgical Oncology*, 2021, 19(1): 337.
- [22] Hu Y, Ping Y, Li L, et al. A retrospective study of risk factors for carbapenem-resistant
klebsiella pneumoniae acquisition among ICU patients[J]. *The Journal of Infection in*
*Developing Countries*, 2016, 10(03): 208-213.
- [23] Al Hamdan A, Alghamdi A, Alyousif G, et al. Evaluating the prevalence and the risk factors of
gram-negative multi-drug resistant bacteria in eastern saudi arabia[J]. *Infection and Drug*
*Resistance*, 2022, Volume 15: 475-490.
- [24] Dao T T, Liebenthal D, Tran T K, et al. Klebsiella pneumoniae oropharyngeal carriage in rural
and urban vietnam and the effect of alcohol consumption[J]. *PLoS ONE*, 2014, 9(3): e91999.
- [25] Xiao T, Zhu Y, Zhang S, et al. A retrospective analysis of risk factors and outcomes of
carbapenem-resistant klebsiella pneumoniae bacteremia in nontransplant patients[J]. *The*
*Journal of Infectious Diseases*, 2020, 221(Supplement_2): S174-S183.
- [26] Li Y, Li J, Hu T, et al. Five-year change of prevalence and risk factors for infection and
mortality of carbapenem-resistant klebsiella pneumoniae bloodstream infection in a tertiary
hospital in north China[J]. *Antimicrobial Resistance & Infection Control*, 2020, 9(1): 79.
- [27] Mehta A C, Muscarella L F. Bronchoscope-related “superbug” infections[J]. *Chest*, 2020,
157(2): 454-469.

- [28] Ponyon J, Kerdsin A, Preeprem T, et al. Risk factors of infections due to multidrug-resistant
gram-negative bacteria in a community hospital in rural thailand[J]. *Tropical Medicine and*
*Infectious Disease*, 2022, 7(11): 328.
- [29] Alicino C, Giacobbe D R, Orsi A, et al. Trends in the annual incidence of carbapenem-resistant
klebsiella pneumoniae bloodstream infections: A 8-year retrospective study in a large teaching
hospital in northern italy[J]. *BMC Infectious Diseases*, 2015, 15(1): 415.
- [30] Kim D, Park B Y, Choi M H, et al. Antimicrobial resistance and virulence factors of klebsiella
pneumoniae affecting 30 day mortality in patients with bloodstream infection[J]. *Journal of*
*Antimicrobial Chemotherapy*, 2019, 74(1): 190-199.
- [31] Choi Y J, Park J Y, Lee H S, et al. Variable effects of underlying diseases on the prognosis of
patients with COVID-19[J]. *PLOS ONE*, 2021, 16(7): e0254258.
- [32] Nilav A, Karimi Rouzbahani A, Mahmoudvand G, et al. Evaluation of the effect of underlying
diseases on mortality of COVID-19 patients: A study of 19,985 cases[J]. *Jundishapur Journal of*
*Microbiology*, 2023, 15(11): e133603.
- [33] Zhang J, Zhao Q, Liu S, et al. Clinical predictive value of the CRP-albumin-lymphocyte index
for prognosis of critically ill patients with sepsis in intensive care unit: A retrospective
single-center observational study[J]. *Frontiers in Public Health*, 2024, 12: 1395134.
- [34] Yang M, Lin S Q, Liu X Y, et al. Association between C-reactive protein-albumin-lymphocyte
(CALLY) index and overall survival in patients with colorectal cancer: From the investigation
on nutrition status and clinical outcome of common cancers study[J]. *Frontiers in Immunology*,
2023, 14: 1131496.
- [35] Cireșă A, Tălăpan D, Vasile C C, et al. Evolution of antimicrobial resistance in klebsiella
pneumoniae over 3 years (2019–2021) in a tertiary hospital in bucharest, romania[J].
*Antibiotics*, 2024, 13(5): 431.
- [36] Wang N, Zhan M, Wang T, et al. Long term characteristics of clinical distribution and resistance
trends of carbapenem-resistant and extended-spectrum β -lactamase klebsiella pneumoniae
infections: 2014–2022[J]. *Infection and Drug Resistance*, 2023, Volume 16: 1279-1295.
- [37] Zhang J, Li D, Huang X, et al. The distribution of K. pneumoniae in different specimen sources
and its antibiotic resistance trends in sichuan, China from 2017 to 2020[J]. *Frontiers in*
*Medicine*, 2022, 9: 759214.
- [38] Falcone M, Russo A, Iacovelli A, et al. Predictors of outcome in ICU patients with septic shock
caused by klebsiella pneumoniae carbapenemase-producing K. pneumoniae[J]. *Clinical*
*Microbiology and Infection*, 2016, 22(5): 444-450.
- [39] Herzig S J, Howell M D, Ngo L H, et al. Acid-suppressive medication use and the risk for
hospital-acquired pneumonia[J]. *JAMA*, 2009, 301(20): 2120-8.
- [40] Laheij, R. J., Sturkenboom, M. C., Hassing, R. J., Dieleman, J., Stricker, B. H., and Jansen, J. B.
Risk of Community-Acquired Pneumonia and Use of Gastric Acid-Suppressive Drugs[Z].
*JAMA*, 2004, 292(16): 1955-1960.
- [41] Savarino V, Di Mario F, Scarpignato C. Proton pump inhibitors in GORDAn overview of their
pharmacology, efficacy and safety[J]. *Pharmacological Research*, 2009, 59(3): 135-153.

Reviewer #1 (Comments for the Author):

This is an interesting study that seems suitable for publication in mSystems. However, I believe the paper requires further revisions before it can be accepted. Below are my suggestions:

--Question 1.1 Ethical Approval: Human studies typically require approval from the Institutional Review Board (IRB) or the Institutional Animal Care and Use Committee (IACUC). The authors should verify if this is applicable.

--Response 1.1: Thank you for your insightful comment and your attention to detail in reviewing our manuscript. We attach great importance to the ethical compliance of the study and have verified the relevant aspects in detail. In fact, the Ethics Committee of the Affiliated Chaohu Hospital of Anhui Medical University has given appropriate ethical review and approval for this study (approval no. KYXM-202312-052). The data utilized in this study was obtained from the hospital's electronic medical record system and did not involve any privacy or information of the patients. Ethical compliance is pivotal to medical research. We have scrupulously executed necessary measures to guarantee the ethical integrity of our study. If required, further approval documents and more ethical details could be provided. **(Page 6, Lines 236-241)**

--Question 1.2 Microorganism Names: The names of microorganisms, such as Pseudomonas aeruginosa ATCC 27853 and Escherichia coli ATCC, as well as Klebsiella pneumoniae, should be italicized. Additionally, the full name should be provided at first mention, followed by abbreviations in subsequent mentions.

--Response 1.2: Thank you for your professional comment, we have revised the relevant sections of the manuscript in response to your comments. **(Page 5, Lines 188; Page 6, Lines 234; Page 7, Lines 281-282)**

--Question 1.3 Table 1 Formatting: There are inconsistencies in capitalization, such as "Respiratory Disease" versus "Cerebrovascular disease." Please ensure consistency throughout the table.

--Response 1.3: Thank you for your valuable suggestion, we have ensured consistency throughout the tables based on your comments. **(Page 9, Table 1; Page 11, Table 2)**

--Question 1.4 Currency Specification in Table 1: If the authors are referring to United States Dollars, this should be explicitly stated, e.g., "United States Dollars" or "US Dollars," as many other countries also use "dollars" (e.g., Singapore Dollars).

--Response 1.4: Thank you for your insightful comment. This study was expressed in US dollars and has been clarified accordingly in the tables. **(Page 10, Table 1; Page 11, Table 2)**

--Question 1.5 Figure 5 Explanation: The numbers 0, 1, 2, 3, 4, 5, and 6 in the figure require clarification. The authors could briefly explain their meaning in the figure caption, such as whether larger numbers indicate higher or lower values, to assist readers in understanding the figure.

--Response 1.5: Thank you for your insightful comment. We have briefly explained the meaning of the horizontal coordinate numbers in the caption notes to Figure 5. Numbers on the horizontal axis represented Odds Ratio (OR), which were used to assess the strength of association between exposure factors and outcomes; a higher value reflected a greater risk of the outcome linked to the exposure factor. **(Figure 5)**

Reviewer #2 (Comments for the Author):

The manuscript presents a comprehensive investigation into the epidemiology, risk factors, clinical outcomes, and predictive modeling for multidrug-resistant *Klebsiella pneumoniae* (MDR-KP) and carbapenem-resistant *Klebsiella pneumoniae* (CRKP) infections at a tertiary hospital in China. The research is timely and important, given the increasing global prevalence of MDR organisms. The study employs robust statistical methods and clearly articulates the findings. However, there are several aspects that could be improved upon.

Major comment:

--Question 2.1: The methodology is generally well-described; however, more specific details regarding the patients' selection criteria for inclusion and exclusion would enhance the reproducibility of the study.

--Response 2.1: Thank you for your constructive feedback regarding the patients' selection criteria for inclusion and exclusion, detailed criteria have been added in the corresponding section. **(Page 4, Lines 133-137)**

--Question 2.2: Clarify the statistical methods used for logistic regression analysis. For instance, provide more details on how the variables were selected for inclusion and the rationale behind this process.

--Response 2.2: Thank you for your thoughtful comment on the statistical methods used to further clarify the logistic regression analyses. We have revised the statistical analysis section to provide more details on how the variables were selected for inclusion and the rationale behind this process. Firstly, variables with $P < 0.05$ in univariate analysis were preliminarily considered to be included in multivariate logistic regression. Subsequently, we undertook a multicollinearity diagnostic analysis. Variables that exhibited variance inflation factor (VIF) > 10 were excluded from analysis in order to mitigate multicollinearity-driven coefficient instability. Given that age and gender were often considered confounders in studies related to bacterial infections, we forced the use of age and gender variables to mitigate confounding biases associated with demographic characteristics in the multivariate logistic regression model. **(Page 6; Lines 249-255)**

Minor Comments:

-Question 2.3 Grammar and Language: Review the manuscript for grammatical errors and sentence structure. For example, certain phrases can be simplified for clarity.

--Response 2.3: Thank you for your valuable comment. We have corrected the grammatical errors and sentence structure in the original manuscript.

Question 2.4 Ethical Considerations: It is mentioned that the study was approved by an ethics board, but more context on obtaining patient consent and data handling in line with ethical practices.

--Response 2.4: Thank you for your professional comment on the ethical aspects of the research. Ethical approval was granted by the Institutional Review Board of the Affiliated Chaohu Hospital of Anhui Medical University (approval no. KYXM-202312-052). The data utilized in this study was obtained from the hospital's electronic medical record system. The research process fully adhered to ethical standards, including the following contents: a) de-identification and anonymization of the data ensured that patient privacy and personal information were fully protected and b) the study did not involve direct intervention with patients. Given the retrospective nature of the study and adherence to the Declaration of Helsinki, the requirement for informed consent was waived by the Ethics Committee. Ethical compliance is pivotal to medical research. We have scrupulously executed necessary measures to guarantee the ethical integrity of our study. If required, further approval documents and more ethical details could be provided. **(Page 6, Lines 236-241)**

Re: mSystems01757-24R1 (Multidrug-resistant *Klebsiella pneumoniae* Coinfection with Multiple Microbes: A Retrospective Study on Its Risk Factors and Clinical Outcomes)

Dear Miss Xixi Song:

Your manuscript has been accepted, and I am forwarding it to the ASM production staff for publication. Your paper will first be checked to make sure all elements meet the technical requirements. ASM staff will contact you if anything needs to be revised before copyediting and production can begin. Otherwise, you will be notified when your proofs are ready to be viewed.

Sincerely,
Naama Geva-Zatorsky